# EquiAV: Single-modal Equivariance Promotes Audio-Visual Contrastive Learning

## Abstract

Advancements in audio-visual representation learning have showcased its effectiveness in acquiring rich and comprehensive representations by leveraging both auditory and visual modalities. Recent works have attempted to improve performance using contrastive learning or masked modeling techniques. However, the effort to maximize the impact of data augmentations for learning semantically rich representation has remained relatively narrow. Without a proper strategy for utilizing data augmentation, the model can be adversely affected or fail to achieve sufficient performance gains. To address this limitation, we present EquiAV, a novel framework that integrates single-modal equivariant contrastive learning with audio-visual contrastive learning. In the proposed framework, audio-visual correspondence and rich modality-specific representations are learned in separate latent spaces. In particular, augmentation-related and modality-specific information is learned in the intra-modal latent space by making the representations equivariant to data augmentation. Extensive ablation studies verify that our framework is the most suitable architecture for maximizing the benefits of the augmentation while ensuring model robustness to strong augmentation. EquiAV outperforms the existing audio-visual self-supervised pre-training methods on audio-visual event classification and zero-shot audio-visual retrieval tasks.

## 1 Introduction

Audio and visual modalities play an important role in how humans perceive their surroundings. Despite the differences in their properties, there exists a natural correspondence between the two modalities. Learning such audio-visual correspondence from large-scale unlabeled video data in a self-supervised manner has recently become a major focus in the deep-learning research community. Among numerous approaches for audio-visual self-supervised learning, *Audio-Visual Contrastive Learning* has been popular due to its simplicity and effectiveness (Gong et al., 2022b; Ma et al., 2021a;b; Morgado et al., 2021a;b; Patrick et al., 2021; Recasens et al., 2021; Wang et al., 2021). This approach primarily learns coordinated representations by encoding separate representations from audio and visual modalities, and then imposing the constraint that the representations of audio-visual pairs from the same video should be closer than those of mismatched samples.

A crucial challenge in audio-visual contrastive learning lies in effectively enhancing representation capability and diversity while preserving the correspondence between two different modalities. The simplest way to enrich representation is to utilize data augmentation. However, employing augmentations in multi-modal contrastive learning requires careful consideration, as augmentations can severely distort the inter-modal correspondence. Recent works propose the methods (Afouras et al., 2022; Chen et al., 2021; Gong et al., 2022b; Ma et al., 2021a;b; Mercea et al., 2022; Mittal et al., 2022; Monfort et al., 2021; Shi et al., 2022; Zhang et al., 2021) to utilize auxiliary tasks to enable the learning of meaningful representation for audio-visual contrastive learning, including masked data modeling and single-modal contrastive learning with self-supervision. Nevertheless, they can adversely affect learning or make the training process unstable when the augmentations are intensely applied. This adverse effect is demonstrated in Figure 1 by the performance of InvAV, which indicates training with single-modal invariant contrastive learning. On the other hand, the model cannot benefit sufficiently from auxiliary tasks by applying weak augmentations.

Meanwhile, numerous single-modal self-supervised learning methods (Bardes et al., 2021; Chen et al., 2020b; Chen & He, 2021; Grill et al., 2020; He et al., 2020; Zbontar et al., 2021) have demonstrated their effectiveness across a wide range of downstream tasks. Many of these methods focus on learning representations that are invariant to augmentations. However, there have been recent advancements in learning equivariant representations (Dangovski et al., 2022; Devillers & Lefort, 2023; Park et al., 2022; Shakerinava et al., 2022), which enable the utilization of powerful augmentations. Equivariant latent space learns to capture augmentation-related information, thereby enhancing the representation capacity.

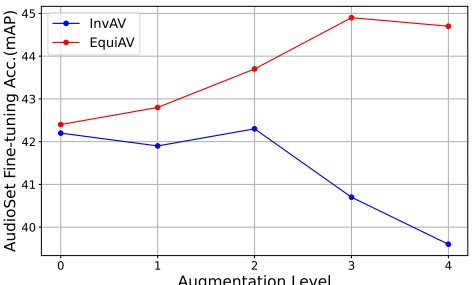

Figure 1: Performance of EquiAV compared to utilizing single-modal invariant contrastive learning as the previous method. A higher augmentation level denotes increasing the variety of augmentations applied. For detailed augmentation settings, refer to Table 4.

In this paper, we propose EquiAV, a framework that integrates single-modal equivariant contrastive learning with conventional audio-visual contrastive learning. The proposed framework leverages separate latent spaces: inter-modal latent space and intra-modal latent space. The model learns coordinated representations that capture audio-visual correspondence in the inter-modal latent space. Modality-unique representations as well as augmentation-related information are learned in the intra-modal latent space using equivariant contrastive loss. EquiAV ensures the full benefit of diverse augmentations applied to both audio and visual modalities. It also demonstrates robustness to distortion of audio-visual correspondence caused by overly strong augmentations as Figure 1.

To the best of our knowledge, we are the first to introduce equivariant contrastive learning to audio-visual self-supervised learning. We investigate the optimal framework for adopting equivariance to the audio-visual scenarios through extensive ablation studies. EquiAV outperforms the existing state-of-the-art audio-visual self-supervised pre-training methods in diverse downstream tasks, including audio-visual event classification and zero-shot audio-visual retrieval tasks. Extensive experimental results validate the effectiveness of EquiAV in learning audio-visual correspondence and enhancing representation capability through the utilization of single-modal equivariant learning. The contribution of our paper is summarised as follows:

- We demonstrate that making the intra-modal representations equivariant to the augmentations shows the most effective method of utilizing data augmentations for audio-visual self-supervised learning.

- We propose EquiAV, an optimal framework that extends the single-modal equivariant representation learning to audio-visual representation learning. It is verified through extensive ablation studies.

- EquiAV outperforms the existing audio-visual self-supervised learning methods in audio-visual event classification and zero-shot audio-visual retrieval tasks. This verifies that EquiAV is an effective framework for learning audio-visual correspondence and semantically rich representation in a self-supervised manner.

## 2 APPROACH

One prominent approach in audio-visual representation learning is to minimize the distance of paired data in the feature space as Figure 2a. This approach ensures that the learned representations align across modalities, facilitating effective multi-modal fusion and enabling tasks such as cross-modal retrieval and audio-visual classification. Given an input pair $(\mathbf{x}_a, \mathbf{x}_v)$ and the encoders for each modality denoted as $f_a$ and $f_v$ respectively, the formulation can be expressed as

$$\min_{f_a, f_v} \mathcal{L}(f_a(\mathbf{x}_a), f_v(\mathbf{x}_v)), \tag{1}$$

where the function $\mathcal{L}(\cdot, \cdot)$ measures the dissimilarity between two inputs. To enhance the generalization capabilities of audio-visual representation learning, various augmentation techniques are employed.

Figure 2: Illustration of the four variants of audio-visual representation learning: (a) conventional audio-visual contrastive learning, (b) audio-visual contrastive learning with augmented inputs only, (c) audio-visual inter-modal contrastive learning with invariant intra-modal self-supervision, and (d) audio-visual inter-modal contrastive learning with equivariant intra-modality self-supervision. $L_{inter}$ and $L_{intra}$ are the terms calculating the similarity of inter- and intra-modality respectively.

## 2.1 Augmentation in Audio-Visual Representation Learning

One approach to incorporate augmentation is to apply it directly to the input pairs as Figure 2b. Then the model captures and encodes the augmented variations, resulting in enhanced representation ability. Given the augmentation distributions $\mathcal{T}_a$ and $\mathcal{T}_v$ applicable to each modality, the objective function can be described as

$$\forall t_a \in \mathcal{T}_a, \forall t_v \in \mathcal{T}_v \quad \min_{f_a, f_v} \mathcal{L}(f_a(t_a(\mathbf{x}_a)), f_v(t_v(\mathbf{x}_v))). \tag{2}$$

However, selecting suitable augmentation distributions poses a challenge as it might break correspondence between the input pairs. Achieving the right balance between augmentation techniques that enhance diversity without distorting the underlying semantic information is a complex and non-trivial task in training with multi-modal inputs.

## 2.2 Intra-modal Self-supervision: Invariance vs Equivariance

In this subsection, we will discuss how to apply intra-modal self-supervision to inter-modal representation learning. By solely applying augmentation within the intra-modal learning process and not extending it to inter-modal matching, we can mitigate the misalignment problem while still reaping the benefits of augmentation.

**InvAV.** One effective approach for providing intra-modal supervision is to utilize the concept of *invariance*(Figure 2c). It targets to align the representations of the original input $\mathbf{x}$ and its augmented counterpart $t(\mathbf{x})$ in the feature space. A number of studies have combined these approaches into audio-visual contrastive learning. We configure this setting as our baseline, named *InvAV*, which can be described as

$$\forall t_a \in \mathcal{T}_a, \forall t_v \in \mathcal{T}_v \quad \min_{f_a, f_v} \mathcal{L}(f_a(\mathbf{x}_a), f_v(\mathbf{x}_v)) + \sum_{a,v} \mathcal{L}(f(\mathbf{x}), f(t(\mathbf{x}))). \tag{3}$$

**EquiAV.** On the other hand, some recent works (Dangovski et al., 2022; Devillers & Lefort, 2023) have suggested that considering the discrepancy of intra-modal pairs in the feature space can lead to better representation learning. This concept, known as *equivariance* (Figure 2d), aims to capture the variations between the original input and its augmented version in a way that preserves the underlying structure. Through this approach, the model can learn to encode not only the invariant features but also the specific transformations applied to each modality. To implement equivariance, an augmentation predictor $u$ along with augmentation parameter $\mathbf{t}$ are employed to match the intra-modal pair in the feature space. Here, $\mathbf{t}$ consists of parameters used to generate the transformation operation $t$. We present the approach, which combines single-modal equivariant contrastive learning with audio-visual contrastive learning, *EquiAV*, and it can be represented as follows:

$$\forall t_a \in \mathcal{T}_a, \forall t_v \in \mathcal{T}_v \quad \min_{u, f_a, f_v} \mathcal{L}(f_a(\mathbf{x}_a), f_v(\mathbf{x}_v)) + \sum_{a,v} \mathcal{L}(u(\mathbf{t}, f(\mathbf{x})), f(t(\mathbf{x}))). \tag{4}$$

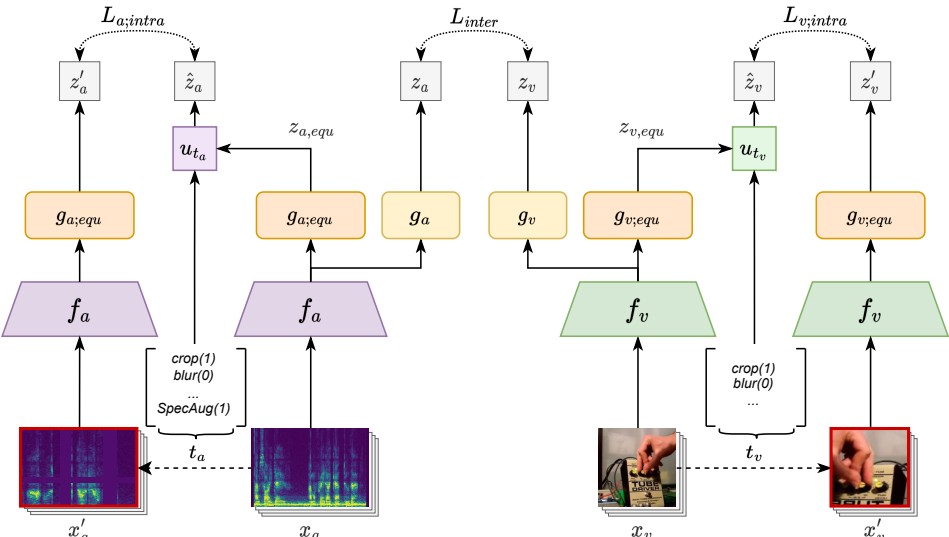

Figure 3: Overview of the proposed EquiAV framework. Audio-visual correspondence and modality-specific representations including augmentation-related information are learned in separate latent spaces using different projection heads.

## 3 METHODS

The proposed framework (Figure 3) consists of (1) an audio encoder $f_a$ and a visual encoder $f_v$, which encode audio-visual inputs and their augmented versions into audio and visual representations, (2) two pairs of heads for each modality that project the audio-visual representations into intra-modal latent space ($g_{a;\text{equ}}$, $g_{v;\text{equ}}$) and inter-modal latent space ($g_a$, $g_v$), and (3) augmentation predictors $u_{t_a}$ and $u_{t_v}$ that predict the displacement within the intra-modal latent spaces caused by the data augmentations. The model is trained with a combination of intra-modal and inter-modal contrastive losses. Each component of our method is explained in detail in the following sections.

### 3.1 EQUIAV

**Data Augmentations and Their Parameterization.** In our framework, typical visual-based augmentations such as random resized crop, color jitter, and horizontal flip are employed for visual inputs. However, unlike the abundant augmentations available for an image or video, the options for the audio modality are relatively limited. To address this, we extend the range of audio augmentation by applying various visual-based augmentations to audio spectrograms. As audio-visual correspondence and augmentation-related information are learned in separate latent spaces, diverse augmentations can be applied without disturbing the inter-modal correspondence. The augmentation information is encoded into real vectors, denoted as $t_a$ and $t_v$. These vectors parameterize how much each augmentation is applied to the input data. They are then concatenated with the audio and visual embeddings before being fed to the augmentation predictors (i.e. $u_{t_a}$, $u_{t_v}$) for intra-modal equivariant representation learning. A detailed explanation of data augmentation parameterization is provided in Appendix B.

**Architecture.** Given the audio-visual inputs and their augmented counterparts, ($x_a$, $x_v$) and ($x'_a$, $x'_v$), the audio encoder $f_a$ and visual encoder $f_v$ encode the inputs into audio and visual representations. Then the inter-modal heads project the audio and visual representations of original inputs into inter-modal latent space to get the embeddings,

$$z_a = g_a(f_a(x_a)), \quad z_v = g_v(f_v(x_v)). \tag{5}$$

Similarly, the intra-modal heads project the representations obtained from the original inputs and their augmented counterparts into intra-modal latent spaces for each modality as follows,

$$z_{a,equ} = g_{a;equ}(f_a(x_a)), \quad z'_a = g_{a;equ}(f_a(x'_a)) \tag{6}$$

$$z_{v,equ} = g_{v;equ}(f_v(x_v)), \quad z'_v = g_{v;equ}(f_v(x'_v)). \tag{7}$$

To model the displacement in the latent space caused by data augmentations, augmentation predictors take as input the concatenation of the original input embeddings and the augmentation vectors, and output equivariant embeddings,

$$\hat{z}_a = u_{t_a}(t_a, z_{a,equ}), \quad \hat{z}_v = u_{t_v}(t_v, z_{v,equ}). \tag{8}$$

## 3.2 LOSS FUNCTIONS

In the training phase for EquiAV, we perform representation learning by maximizing the similarity between the specified positive pairs in both intra-modal and inter-modal settings. To avoid the collapse of all representations towards a constant value, we employ contrastive learning by designating negative pairs. The contrastive loss is handled within each subspace by calculating the similarity between audio-audio, visual-visual, and audio-visual pairs. Additionally, cosine similarity is used to measure the similarity between the pair.

**Intra-modal Representation Learning.** Equivariant contrastive learning is used for training intra-domain pairs. The batch consists of $N$ equivariant embeddings $\{z'_i\}_{i \in \{1,...,N\}}$ of the augmented inputs, as well as $N$ embeddings $\{\hat{z}_i\}_{i \in \{1,...,N\}}$ obtained by incorporating the augmentation information and original input. The pair $(z'_i, \hat{z}_i)$ generated from the same image forms a positive pair, while the remaining $2(N-1)$ embeddings within the batch serve as negative pairs. The NT-Xent loss (Chen et al., 2020b) is employed to calculate the contrastive loss. The loss for equivariant embeddings of the augmented input can be expressed as

$$\ell_{intra}(\hat{z}, z') = -\frac{1}{N} \sum_{i=1}^{N} \log \frac{\exp(\text{sim}(\hat{z}_i, z'_i)/\tau)}{\sum_{j=1}^{N} \exp(\text{sim}(\hat{z}_i, z'_j)/\tau) + \sum_{j=1}^{N} \mathbb{1}_{[j \neq i]} \exp(\text{sim}(\hat{z}_i, \hat{z}_j)/\tau)}, \tag{9}$$

where $\tau > 0$ is temperature, and $\mathbb{1}$ is an indicator function that assigns a value of 1 or 0 to an element based on whether it satisfies a given condition. Then, the intra-modal loss is described by combining the losses of adopting the original and augmented inputs as anchors respectively,

$$\mathcal{L}_{intra} = \frac{1}{2}(\ell_{intra}(\hat{z}, z') + \ell_{intra}(z', \hat{z})). \tag{10}$$

Note that the equivariance loss term in our framework differs slightly from the single-modal equivariant self-supervised learning (Devillers & Lefort, 2023). The key distinction lies in whether the similarity of the positive pair is included in the denominator of the loss term or not. Detailed explanations are described in the Appendix A.

**Inter-modal Representation Learning.** On the other hand, we apply invariant contrastive learning for training inter-domain pairs. The batch consists of $N$ paired embeddings $\{(z_i^a, z_i^v)\}_{i \in \{1,...,N\}}$, which are extracted without applying any augmentation to preserve the correspondence between input pairs. In the process of calculating contrastive loss, each anchor utilizes embeddings from other domains, excluding its domain. For example, considering an anchor $z_i^a$ from the audio domain, the positive pair will be $(z_i^a, z_i^v)$, while the remaining visual embeddings within the batch would serve as negative pairs. Accordingly, the loss for the audio domain and visual domain can be expressed as

$$\ell_{inter}(z^a, z^v) = -\frac{1}{N} \sum_{i=1}^{N} \log \frac{\exp(\text{sim}(z_i^a, z_i^v)/\tau)}{\sum_{j=1}^{N} \exp(\text{sim}(z_i^a, z_j^v)/\tau)}, \tag{11}$$

where $\tau > 0$ is temperature. Then, the inter-modal loss is represented by combining the losses of adopting the audio and visual embeddings as anchors respectively,

$$\mathcal{L}_{inter} = \frac{1}{2}(\ell_{inter}(z^a, z^v) + \ell_{inter}(z^v, z^a)). \tag{12}$$

Table 1: Audio-visual event classification performance on AudioSet and VGGSound. **A**: Audio-only, **V**: Visual-only, **A-V**: Audio-visual. IN SL: ImageNet supervised learning, SSL: Self-supervised learning, $^\dagger$ Non-standard train/test split. We de-emphasize concurrent work.

| Method | Pretrain | AudioSet-20K (mAP) | | | AudioSet-2M (mAP) | | | VGGSound (Acc) | | |
|---|---|---|---|---|---|---|---|---|---|---|
| | | A | V | A-V | A | V | A-V | A | V | A-V |
| *Audio-Based Models* | | | | | | | | | | |
| PANNs (Kong et al., 2020) | - | 27.8 | - | - | 43.9 | - | - | - | - | - |
| AST (Gong et al., 2021) | IN SL | 34.7 | - | - | 45.9 | - | - | - | - | - |
| HTS-AT (Chen et al., 2022) | IN SL | - | - | - | 47.1 | - | - | - | - | - |
| PaSST (Koutini et al., 2021) | IN SL | - | - | - | 47.1 | - | - | - | - | - |
| SSAST (Gong et al., 2022a) | SSL | 31.0 | - | - | - | - | - | | | |
| MAE-AST (Baade et al., 2022) | SSL | 30.6 | - | - | - | - | - | | | |
| Audio-MAE (Huang et al., 2022b) | SSL | 37.1 | - | - | 47.3 | - | - | - | - | - |
| AudioSlowFast (Kazakos et al., 2021) | - | - | - | - | - | - | - | 52.5 | - | - |
| *Audio-Visual Based Models* | | | | | | | | | | |
| GBlend (Wang et al., 2020) | - | 29.1 | 22.1 | 37.8 | 32.4 | 18.8 | 41.8 | - | - | - |
| Perceiver (Jaegle et al., 2021) | - | - | - | - | 38.4 | 25.8 | 44.2 | - | - | - |
| Attn AV (Fayek & Kumar, 2020) | IN SL | - | - | - | 38.4 | 25.7 | 46.2 | - | - | - |
| MBT$^\dagger$ (Nagrani et al., 2021) | IN21K SL | 31.3 | **27.7** | 43.9 | 41.5 | **31.3** | 49.6 | 52.3 | **51.2** | 64.1 |
| CAV-MAE (Gong et al., 2022b) | SSL | 37.7 | 19.8 | 42.0 | 46.6 | 26.2 | 51.2 | 59.5 | 47.0 | 65.5 |
| AudiovisualMAE$^\dagger$ (Georgescu et al., 2022) | SSL | - | - | - | 46.6 | 31.1 | 51.8 | 57.2 | 50.3 | 65.0 |
| MAViL (Huang et al., 2022a) | SSL | 41.8 | 24.8 | 44.9 | 48.7 | 30.3 | 53.3 | 60.8 | 50.9 | 67.1 |
| EquiAV (ours) | SSL | **42.3** | 24.9 | **44.9** | **47.5** | 30.1 | **52.6** | **60.0** | 48.1 | **66.4** |

Finally, the loss function of EquiAV can be expressed by incorporating weighting factors to the intra-modal losses of audio and visual modalities, as well as the audio-visual inter-modal loss. It can be formulated as follows:

$$\mathcal{L}_{EquiAV} = \lambda_{inter} \cdot \mathcal{L}_{inter} + \lambda_{a;intra} \cdot \mathcal{L}_{a;intra} + \lambda_{v;intra} \cdot \mathcal{L}_{v;intra} \tag{13}$$

Through these methods, EquiAV can maximize the benefits of intra-domain supervision while avoiding any detrimental impact on the learning of inter-modal pairs. In the next section, we will provide experimental evidence supporting our design choices along with benchmark performance.

## 4 EXPERIMENTS

Firstly, the model is pre-trained on the AudioSet-2M in a self-supervised fashion, without the use of labels. We evaluate our model through audio-visual event classification in both single-modal and multi-modal manner, as well as zero-shot audio-visual retrieval task. During these tasks, we remove the intra-modal head $g_{equ}$ and utilize the feature obtained by passing through the backbone encoder $f$ and inter-modal head $g$.

### 4.1 MAIN RESULTS

**Implementation Details.** The audio encoder $f_a$ and visual encoder $f_v$ are initialized with the self-supervised pre-trained ViT-B/16 model of MAE (He et al., 2022). We employ linear layers for inter-modal heads while using a 3-layer MLP with layer normalization for intra-modal heads and augmentation predictors. More detailed experimental settings are explained in the Appendix B.

**Audio-Visual Event Classification.** The model is fine-tuned using the AudioSet-20K and VG-GSound datasets, and evaluated on audio-visual event classification to indicate its representation capability. Fine-tuning is performed by adding a linear layer to the representations; for single-modal fine-tuning, the feature of each modality is utilized as the single-modal representation, whereas for multi-modal fine-tuning, the features from both modalities are concatenated to form the joint representation. Table 1 provides a comparison of the single-modal and multi-modal fine-tuning performance. EquiAV surpasses the performance of previous methods by solely relying on contrastive learning. Note that the performance of MBT (Nagrani et al., 2021) and MAViL (Huang et al., 2022a) is the most recently reported performance.

Table 2: Zero-shot audio-visual retrieval results on the AudioSet and VGGSound with models pre-trained on AudioSet-2M. [†] Results reproduced on our environment.[1]

| Method | Video-to-Audio | | | Audio-to-Video | | |
|---|---|---|---|---|---|---|
| | R@1 | R@5 | R@10 | R@1 | R@5 | R@10 |
| *AudioSet* | | | | | | |
| CAV-MAE-Base[†] | 10.2 | 26.8 | 35.6 | 7.0 | 18.0 | 26.0 |
| CAV-MAE-Scale+[†] | 14.2 | 32.4 | 41.3 | 10.9 | 26.2 | 34.8 |
| CAV-MAE-Scale++[†] | 16.6 | 37.0 | 45.9 | 14.3 | 32.0 | 40.7 |
| CAV-MAE (Gong et al., 2022b) | 18.8 | 39.5 | 50.1 | 15.1 | 34.0 | 43.0 |
| EquiAV (ours) | **27.7** | **51.2** | **60.0** | **25.9** | **50.2** | **56.6** |
| *VGGSound* | | | | | | |
| CAV-MAE-Base[†] | 11.8 | 27.0 | 36.8 | 8.5 | 22.5 | 30.9 |
| CAV-MAE-Scale+[†] | 13.1 | 31.0 | 41.4 | 12.0 | 29.6 | 37.9 |
| CAV-MAE-Scale++[†] | 15.5 | 35.3 | 45.1 | 16.4 | 35.0 | 44.7 |
| CAV-MAE (Gong et al., 2022b) | 14.8 | 34.2 | 44.0 | 12.8 | 30.4 | 40.3 |
| EquiAV (ours) | **24.4** | **47.3** | **56.0** | **23.7** | **46.6** | **57.1** |

**Zero-Shot Audio-Visual Retrieval.** We evaluate the generalizability of the model through the zero-shot audio-visual retrieval task. The retrieval task was performed based on the similarity between the features of each modality. Table 2 presents the results of a zero-shot retrieval task on the subset of the evaluation samples from AudioSet and VGGSound. The sample list used for the experiments is identical to the one used in CAV-MAE (Gong et al., 2022b). Additionally, to demonstrate the robustness of our model across different datasets, we perform further zero-shot retrieval experiments on MSR-VTT (Xu et al., 2016) in Appendix C.1.

Our evaluation results have demonstrated the enhanced representation capability and generalizability of our proposed method, as shown by the superior performance on the audio-visual event classification and zero-shot retrieval tasks.

## 4.2 ABLATION STUDIES

In this section, a series of experiments are conducted to identify the optimal framework for leveraging equivariance in the context of audio-visual representation learning. First, we determine that EquiAV, which utilizes intra-modal equivariance, is the most suitable framework for learning audio-visual correspondence and joint representations. The findings further show that EquiAV maximizes the benefits of data augmentations applied to multi-modal inputs while showing the robustness to excessive augmentations which may impair audio-visual correspondence. Lastly, we demonstrate the effectiveness of the employed loss function for intra-modal equivariant contrastive learning.

Table 3: Zero-shot retrieval results on AudioSet and audio-visual event classification performance on AudioSet-20K with the variants of pre-training methods, including those introduced in Figure 2. **V2A**: Video-to-audio zero shot retrieval R@1, **A2V**: Audio-to-video zero shot retrieval R@1, **Inv**: Invariant contrastive learning, **Equi**: Equivariant contrastive learning, **Aug**: Data augmentation applied.

| Method | Intra-modal | Inter-modal | Zero-shot Retrieval | | Fine-Tuning | | |
|---|---|---|---|---|---|---|---|
| | | | V2A | A2V | A | V | A-V |
| - | - | Inv | 22.0 | 20.2 | 32.0 | 18.4 | 37.0 |
| - | - | Inv (w/ Aug) | 13.4 | 12.6 | 32.1 | 18.7 | 38.8 |
| InvAV | Inv | Inv | 25.2 | 24.3 | 40.1 | 22.9 | 42.3 |
| - | Inv | Equi | 14.3 | 13.1 | 37.2 | 20.4 | 41.9 |
| EquiAV | Equi | Inv | **27.7** | **25.9** | **42.3** | **24.9** | **44.9** |

**Audio-Visual Representation Learning Frameworks.** We implement a variety of distinct pre-training strategies to demonstrate which framework most effectively extracts comprehensive information from multi-modal inputs. As presented in Table 3, focusing solely on inter-modal representation without augmentations (illustrated in Figure 2a) results in the lowest fine-tuning task scores. This

---

[1]Weights from https://github.com/YuanGongND/cav-mae

Table 4: Zero-shot retrieval results on AudioSet and audio-visual event classification performance on AudioSet-20K with varying augmentations.

| Method | Visual Augmentation | Audio Augmentation | Zero-shot Retrieval | | Fine-Tuning | | |
|---|---|---|---|---|---|---|---|
| | | | V2A | A2V | A | V | A-V |
| EquiAV | RRC+CJ+GB | SA+TS | 25.0 | 24.6 | 39.9 | 22.4 | 42.4 |
| | RRC+CJ+GB+HF+GS | SA+TS | 26.5 | 24.9 | 40.4 | 24.5 | 42.8 |
| | RRC+CJ+GB+HF+GS | SA+TS+RRC+CJ | 26.3 | 25.4 | 41.6 | 24.0 | 43.7 |
| | RRC+CJ+GB+HF+GS | SA+TS+RRC+CJ+GB+HF | **27.7** | **25.9** | **42.3** | **24.9** | **44.9** |
| | RRC+CJ+GB+HF+GS+FR+VF | SA+TS+RRC+CJ+GB+HF | 27.0 | 25.1 | 42.0 | 24.3 | 44.7 |
| InvAV | RRC+CJ+GB | SA+TS | 24.7 | 23.7 | 39.8 | 22.1 | 42.2 |
| | RRC+CJ+GB+HF+GS | SA+TS | 24.6 | 24.0 | 39.4 | 22.7 | 41.9 |
| | RRC+CJ+GB+HF+GS | SA+TS+RRC+CJ | 25.2 | 24.3 | 40.1 | 22.9 | 42.3 |
| | RRC+CJ+GB+HF+GS | SA+TS+RRC+CJ+GB+HF | 22.9 | 22.2 | 38.3 | 22.6 | 40.7 |
| | RRC+CJ+GB+HF+GS+FR+VF | SA+TS+RRC+CJ+GB+HF | 22.7 | 21.9 | 38.0 | 19.8 | 39.6 |

Table 5: Zero-shot retrieval results on AudioSet and audio-visual event classification performance on AudioSet-20K with different loss functions.

| Intra-modal Loss | Zero-shot Retrieval | | Fine-Tuning | | |
|---|---|---|---|---|---|
| | V2A | A2V | A | V | A-V |
| without pos (Eq. A) | 18.7 | 17.4 | 41.0 | 22.8 | 43.4 |
| with pos (Eq. 9) | **27.7** | **25.9** | **42.3** | **24.4** | **44.9** |

suggests that only using original audio and visual inputs is inadequate for acquiring sufficient representation capabilities. In the case of using augmented inputs (illustrated in Figure 2b), the performance of zero-shot retrieval is significantly degraded. This decline in performance is attributed to augmentations that distort the semantic context of the data, thereby adversely affecting the inter-modal correspondence. Additionally, we investigated the effectiveness of incorporating equivariance into audio-visual contrastive learning, specifically analyzing whether its application is more impactful within intra-modal or inter-modal contexts. The results (fourth row in Table 3) show that applying equivariance to inter-modal latent space is not beneficial and may even hinder the learning of audio-visual correspondences and joint representations. Since audio-visual correspondence is the strongest when the semantic context of audio-visual inputs remains unchanged, it can be inferred that the augmentation-related information and audio-visual correspondence should be learned in separate latent spaces. Our results indicate that intra-modal equivariant learning is the most effective strategy for learning inter-modal correspondence and for capturing semantically richer information across different modalities in the context of audio-visual contrastive learning.

**Data Augmentations.** We explore a variety of augmentations applied to visual and audio modalities in the pre-training stage. For the visual modality, the typical visual-based augmentations are used; Random Resized Crop (RRC), Color Jitter (CJ), Gaussian Blur (GB), Horizontal Flip (HF), and Gray Scale (GS). In the case of audio modality, both audio- and visual-based augmentations are applied to the audio spectrogram. Specifically, SpecAugment (SA) (Park et al., 2019) and Time Shifting (TS) are utilized as audio-based augmentations. Meanwhile, the same augmentations pool used for the visual modality excluding GS is applied to audio spectrograms, as the spectrogram has only one channel. As shown in Table 4, EquiAV fully benefits from increasing the variety of the data augmentations. Moreover, EquiAV exhibits higher robustness to excessive augmentations (e.g. Four-fold Rotation (FR) and Vertical Flip (VF)) compared to InvAV. The results imply that by utilizing equivariant intra-modal representation learning, our model is able to capture augmentation-related information within each modality. Consequently, EquiAV aids in learning more robust audio-visual correspondence and joint representations, thus showing promise for handling a wide range of data augmentations.

**Equivariant Loss Functions.** Table 5 shows the results of using different equivariant loss functions for intra-modal pairs. The first row excludes the positive pair in the denominator according to Equation A, while the second row includes it as in Equation 9. When Equation 9 is adopted as the

intra-modal contrastive loss, the weight updates due to hard positives are relatively larger than those due to easy positives. In the context of equivariant contrastive learning, learning from hard positive utilizing the augmentation predictor proves to be more effective, leading to better representation quality. The experimental results support our hypothesis. For further analytical insights, refer to Appendix A.

## 5 RELATED WORKS

**Audio-Visual Representation Learning.** Audio-visual contrastive learning has been one of the most popular approaches for learning the natural correspondence between audio and visual modalities from video, due to its simple intuition and effectiveness (Owens & Efros, 2018; Recasens et al., 2021; Georgescu et al., 2022; Sarkar & Etemad, 2023). One common approach of contrastive learning is to learn the context of the synchronous relationship between audio and visual inputs (Korbar et al., 2016; Alwassel et al., 2020; Morgado et al., 2021b; Sarkar & Etemad, 2023). Other research improves the impact of contrast learning by applying additional data augmentation (Patrick et al., 2021; Wang et al., 2021) or mining harder negatives (Ma et al., 2021a; Morgado et al., 2021a). On the other hand, several works (Georgescu et al., 2022; Haliassos et al., 2022) adopt masked modeling techniques that are designed to reconstruct the masked raw inputs or predict the masked context features. Recently, CAV-MAE (Gong et al., 2022b) and our concurrent work MAViL (Huang et al., 2022a) have similar goals to our work, combining contrastive learning and masked data modeling techniques to learn complementary representations. Our approach instead incorporates single-modal equivariant representation learning with audio-visual contrastive learning, which enables learning of modality-specific representation and augmentation-related information while preserving the audio-visual correspondence. This leads to improved performance in various downstream tasks.

**Single-modal Self-supervised Learning.** Self-supervised learning that leverages large-scale datasets has shown promising performance in various fields. Early research in this domain was characterized by the creation of various handcrafted pretext tasks (Doersch et al., 2015; Gidaris et al., 2018; Noroozi & Favaro, 2016; Pathak et al., 2016; Zhang et al., 2016). More recently, the field has seen a surge in popularity with the advent of techniques such as contrastive learning (Chen et al., 2020b; He et al., 2020; Oord et al., 2018), masked modeling (He et al., 2022; Tong et al., 2022; Huang et al., 2022b), and non-contrastive methods (Caron et al., 2021; Chen & He, 2021; Grill et al., 2020). Contrastive learning is designed to learn invariant semantics by employing an appropriate augmentation pool and treating the augmented data as positive pairs. However, recent studies (Dangovski et al., 2022; Devillers & Lefort, 2023) have indicated that performance can be enhanced by incorporating the principle of equivariance, which allows for displacement in the feature space in response to data augmentations. One such approach is predicting augmentations on the input data (Dangovski et al., 2022). Another strategy involves integrating augmentation information directly into the learning process (Devillers & Lefort, 2023; Lee et al., 2021).

## 6 CONCLUSION

We propose EquiAV that incorporates single-modal equivariant contrastive learning within an audio-visual contrastive learning framework. Making intra-modal representations equivariant to data augmentations has been proven to be an effective approach in exploiting single-modal self-supervision for audio-visual self-supervised learning. However, simply replacing the single-modal invariant representation learning with the equivariant counterpart does not guarantee performance improvement on downstream tasks. By extensive ablation studies for searching the optimal framework, we finally enable the model to learn inter-modal correspondence while enriching modality-specific representation at the same time. Single-modal equivariant learning successfully circumvents the typical constraints on data augmentation strategies and maximizes their benefits in audio-visual self-supervised learning. This can be demonstrated by the improved performance driven by applying stronger augmentations to both modalities. Furthermore, additional benchmark experimental results and various ablation studies can be found in Appendix C. EquiAV outperforms the existing state-of-the-art audio-visual self-supervised learning methods on audio-visual event classification tasks and zero-shot audio-visual retrieval tasks.

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

# A  APPENDIX

## A  ANALYSIS ON EQUIVARIANT LOSS FUNCTIONS

As mentioned in Section 3.2, our equivariant loss function differs from the equivariant contrastive loss function used in EquiMod (Devillers & Lefort, 2023), regarding whether a positive pair similarity is included in the denominator. The equivariant loss of EquiMod is the same as applying an indicator function to both summation terms in the denominators of Equation 9, which can be represented as follows:

$$\ell_{EquiMod}(\hat{z}, z') = -\frac{1}{N} \sum_{i=1}^{N} \log \frac{\exp(\mathrm{sim}(\hat{z}_i, z'_i)/\tau)}{\sum_{j=1}^{N} \mathbb{1}_{[j \neq i]} \left[\exp(\mathrm{sim}(\hat{z}_i, z'_j)/\tau) + \exp(\mathrm{sim}(\hat{z}_i, \hat{z}_j)/\tau)\right]}, \quad \text{(A)}$$

$$\mathcal{L}_{EquiMod} = \frac{1}{2}(\ell_{EquiMod}(\hat{z}, z') + \ell_{EquiMod}(z', \hat{z})). \quad \text{(B)}$$

Consider intra-modal training batch embeddings as $\{z_i\} = \{\hat{z}_i\} \cup \{z'_i\}$. For each $i$-th embedding, let's denote the positive sample as $p_i$ and the set of negative samples as $N_i$. Then, the equivariant loss terms of EquiMod and EquiAV can be simply rewritten as

$$\mathcal{L}_{i;equ}^{EquiMod} = -\log \frac{s_{i,p_i}}{\sum_{n \in N_i} s_{i,n}}, \quad \text{(C)}$$

$$\mathcal{L}_{i;equ}^{EquiAV} = -\log \frac{s_{i,p_i}}{s_{i,p_i} + \sum_{n \in N_i} s_{i,n}}, \quad \text{(D)}$$

where $s_{i,j} = \exp(\mathrm{sim}(z_i, z_j)/\tau)$. Differentiating the above equations with respect to $s_{i,p_i}$ yields the following expressions:

$$\frac{\partial \mathcal{L}_{i;equ}^{EquiMod}}{\partial s_{i,p_i}} = -\frac{1}{s_{i,p_i}}, \quad \text{(E)}$$

$$\frac{\partial \mathcal{L}_{i;equ}^{EquiAV}}{\partial s_{i,p_i}} = -\frac{\sum_{n \in N_i} s_{i,n}}{s_{i,p_i} \left(s_{i,p_i} + \sum_{n \in N_i} s_{i,n}\right)}$$
$$= \frac{\partial \mathcal{L}_{i;equ}^{EquiMod}}{\partial s_{i,p_i}} \cdot \frac{\sum_{n \in N_i} s_{i,n}}{\left(s_{i,p_i} + \sum_{n \in N_i} s_{i,n}\right)} \quad \text{(F)}$$

When we compare the EquiAV loss to the EquiMod loss through Equ.F and Equ. E, the EquiAV loss puts relatively more weight on hard positive compared to easy positive. This becomes particularly advantageous when stronger augmentations lead to an increased frequency of hard positives. Training with more hard positives, alongside the application of equivariance, substantially enhances the model's capability to comprehend detailed features in single-modal contexts. The intra-modal representation quality plays a pivotal role in understanding and integrating different modalities. Consequently, the semantically rich intra-modal representation promotes effective learning of audio-visual correspondence and audio-visual joint representations. Experimental results in Table 5 support our hypothesis.

## B  IMPLEMENTATION DETAILS

**Datasets**  We utilize two prominent audio-visual datasets for our experiments: AudioSet (Gemmeke et al., 2017) and VGGSound (Chen et al., 2020a). AudioSet comprises 2 million 10-second YouTube clips, designed to classify events into 527 distinct classes, with each data having multiple labels. We download 1,893,278 clips for the AudioSet-2M, 21,074 clips for the AudioSet-20K, and 19,446 clips for evaluation. Particularly, AudioSet-20K is a subset of AudioSet-2M. VGGSound includes 200,000 10-second YouTube clips, encompassing 309 classes. The training and test splits of VGGSound

consist of 183,730 and 15,446 downloaded clips, respectively. Unlike AudioSet, it has only one label for each clip. For the zero-shot retrieval evaluation, we collect 1,722 and 1,545 clips from AudioSet and VGGSound's evaluation set respectively.

**Input Pre-processing.** We follow AST (Gong et al., 2021) and ViT (Dosovitskiy et al., 2021) for pre-processing of audio and visual inputs, respectively. For audio, each 10-second audio waveform is transformed into a sequence of 128-dimensional log Mel filterbank features by using a 25-ms Hanning window and a 10-ms hop size, resulting in a 1024(time) $\times$ 128(frequency) spectrogram. For visual inputs, 10 frames are uniformly extracted from each 10-second video, and one frame is randomly selected as the input. Then the audio spectrogram and the video frame are tokenized to 16 $\times$ 16 patches and fed to the audio and visual encoders.

**Augmentation Parameterization.** Audio and visual augmentations as well as their parameterization used in this work are listed as follows:

- *Random resized crop* (4 elements for both audio & visual): $x$ and $y$ coordinates of the top-left point as well as the width and height of the crop (<0, 0, 0, 0> is used as a default encoding).

- *Color jitter* (8 elements for visual & 4 elements for audio): the jitter factors for brightness, contrast, saturation, and hue of video frames, as well as the order of the application of transformation. We use the following mapping to encode the order of transformation: {0: brightness, 1: contrast, 2: saturation, 3: hue}. For instance, an encoding <2, 1, 3, 0> indicates that the saturation jitter is first applied, and then contrast, hue, and brightness (<0, 1, 2, 3> is used as default). On the other hand, we only use brightness and contrast jitters for audio spectrograms, as the audio spectrograms are originally grayscale. Then, we use the following mapping to encode the order of jitter transformation: 0: brightness, 1: contrast (<0, 1> is used as default).

- *Gaussian blur* (1 element for both audio & visual): the value of σ for Gaussian blurring kernel (0 as default).

- *Horizontal flip* (1 element for both audio & visual): 1 if an image or an audio spectrogram is horizontally flipped and 0 otherwise.

- *Grayscale* (1 element, for visual only): 1 if an image is converted to grayscale and 0 otherwise.

- *Random time shifting* (1 element, for audio only): the value of temporal shift of the audio spectrogram.

- *SpecAug* (Park et al., 2019) (4 elements, for audio only): starting and end points of the masking along the time and frequency axis of the audio spectrogram.

All augmentations except random resized crop (which is always applied) are applied with pre-defined probability. Therefore for each augmentation, we add an element, whose value is 1 when the augmentation is actually applied and 0 otherwise, to a parameterized vector. Consequently, the audio and visual augmentations are encoded into 24-dimensional and 18-dimensional vectors respectively. They are then concatenated with the audio and visual representations and projected into a 256-dimensional latent space by the augmentation predictor, which is a 3-layer MLP. The whole model including the augmentation predictor is learned jointly.

## C    ADDITIONAL EXPERIMENTS

### C.1    MAIN RESULTS

**Zero-Shot Audio-Visual Retrieval Experiments.** We also perform a zero-shot retrieval evaluation on the MSR-VTT (Xu et al., 2016) using the model pre-trained on AudioSet-2M. As shown in Table A, our model consistently outperforms existing methods pre-trained on AudioSet and achieves comparable results to models pre-trained on much larger datasets. Note that we use a subset of 1,000 out of the total 2,635 test data for retrieval following previous work when conducting experiments on the test set.

Table A: Zero-shot retrieval on the test and evaluation splits of MSR-VTT. [†] Results reproduced on our environment using the model weights reported in previous work.

| | | Video-to-Audio | | | Audio-to-Video | | |
|---|---|---|---|---|---|---|---|
| Method | Pre-train Dataset | R@1 | R@5 | R@10 | R@1 | R@5 | R@10 |
| *Test set:* | | | | | | | |
| Boggust (Boggust et al., 2019) | HowTo100M | 9.3 | 20.7 | 28.8 | 7.6 | 21.1 | 28.3 |
| Aranjelovic (Arandjelovic & Zisserman, 2018) | HowTo100M | 11.9 | 25.9 | 34.7 | 12.6 | 26.3 | 33.7 |
| AVLnet (Rouditchenko et al., 2021) | HowTo100M | **17.2** | 26.6 | **46.6** | 17.8 | **35.5** | **43.6** |
| CAV-MAE (Gong et al., 2022b) | AudioSet-2M | 13.3 | 29.0 | 40.5 | 7.6 | 19.8 | 30.2 |
| EquiAV (Ours) | AudioSet-2M | 13.6 | **31.4** | 40.7 | 13.5 | 32.6 | 42.0 |
| *Eval set:* | | | | | | | |
| CAV-MAE-Base[†] | AudioSet-2M | 9.8 | 27.0 | 36.5 | 6.7 | 22.8 | 32.6 |
| CAV-MAE-Scale+[†] | AudioSet-2M | 11.1 | 31.7 | 41.5 | 10.7 | 30.1 | 38.8 |
| CAV-MAE-Scale++[†] | AudioSet-2M | 13.7 | 35.4 | 45.2 | 11.8 | 31.9 | 43.3 |
| EquiAV (Ours) | AudioSet-2M | **18.7** | **36.1** | **46.3** | **16.3** | **37.3** | **49.8** |

**Single-modal Downstream Tasks.** We conduct experiments to demonstrate the effectiveness of our method in single-modal downstream tasks. For the video-only task, we utilize the UCF101 (Soomro et al., 2012) and HMDB51 (Kuehne et al., 2011), and for the audio-only task, the ESC-50 (Piczak, 2015). According to the results reported in Table B, our method shows superior performance in the video-only task compared to previous methods and yielded comparable outcomes in the audio-only task.

Table B: Action recognition (video-only) results on UCF101 and HMDB51, and environmental sound classification (audio-only task) results on ESC-50. IN: ImageNet, AS: AudioSet-2M, SL: Supervised learning, SSL: Self-supervised learning. [†]Results reproduced on our environment using the model weights reported in previous work.

| | | Video-only (Acc) | | Audio-only (Acc) |
|---|---|---|---|---|
| Method | Pretrain | UCF101 | HMDB51 | ESC-50 |
| *Audio-Based Models* | | | | |
| PANNs (Kong et al., 2020) | AS-SL | - | - | 94.7 |
| AST (Gong et al., 2021) | IN-SL | - | - | 88.7 |
| AST (Gong et al., 2021) | IN-SL, AS-SL | - | - | 95.6 |
| HTS-AT (Chen et al., 2022) | IN-SL | - | - | **97.0** |
| PaSST (Koutini et al., 2021) | IN-SL | - | - | 96.8 |
| SSAST (Gong et al., 2022a) | AS-SSL | - | - | 88.8 |
| MAE-AST (Baade et al., 2022) | AS-SSL | - | - | 90.0 |
| Audio-MAE (Huang et al., 2022b) | AS-SSL | - | - | 94.1 |
| *Audio-Visual Based Models* | | | | |
| CAV-MAE-Scale+[†] (Gong et al., 2022b) | IN-SSL, AS-SSL | 75.9 | 43.5 | 84.0 |
| MAViL (Huang et al., 2022a) | AS-SSL | - | - | 94.4 |
| MAViL (Huang et al., 2022a) | IN-SSL, AS-SSL | - | - | 94.4 |
| EquiAV (ours) | IN-SSL, AS-SSL | **87.0** | **65.5** | 96.0 |

## C.2 ABLATION STUDIES

**Pre-training Architecture Search.** In these experiments, we incorporate an extra augmentation encoder during the pre-training stage. The role of the augmentation encoder is to encode the augmentation parameter $t_m$ ($\{a, v\} \in m$) into 128-dim feature $\tilde{t}_m$ before feeding it as input to the augmentation predictor $u_{t_m}$. Figure Aa and Figure Ab illustrate the case without and with the augmentation encoder, respectively. According to Table C, the event classification performance without the augmentation encoder outperforms the scores achieved with the augmentation encoder. Moreover, we observe that fine-tuning using the representations from the inter-modal head yields the best performance. This indicates the representations derived from the inter-modal head exhibit superior capabilities for representation learning.

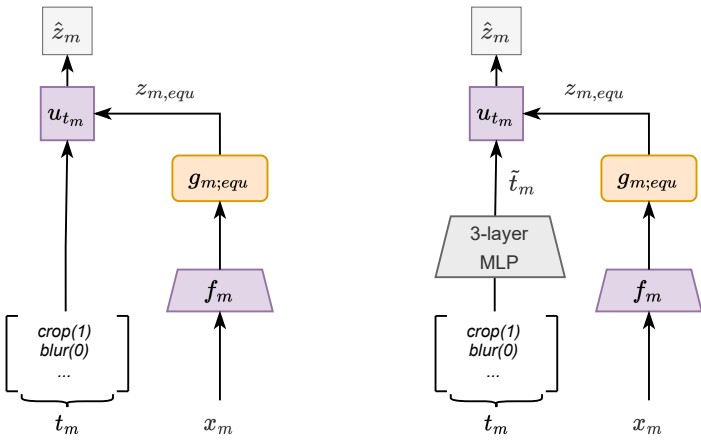

(a) Without augmentation encoder      (b) With augmentation encoder

Figure A: Illustration of using augmentation encoder.

Table C: Event classification performance on AudioSet-20K based on the presence of augmentation encoder and varying head types.

| Aug Encoder | Head | A | V | A-V |
|:---:|:---|:---:|:---:|:---:|
| ✓ | inter-head($g$) | 40.4 | 22.0 | 43.3 |
| ✗ | inter-head($g$) | **42.3** | **24.9** | **44.9** |
| ✗ | intra-head($g_{equ}$) | 34.1 | 17.8 | 37.8 |
| ✗ | no head($f_m(x_m)$) | 32.0 | 6.7 | 35.0 |

**Batch Size.** For a fair comparison with the previous methods in terms of batch size, we conduct additional experiments with the same total batch size. As shown in Table D, our method outperforms the previous method, which supports that our method is sufficiently helpful in learning audio-visual representations, without using a larger batch size.

Table D: Zero-shot retrieval results on AudioSet and audio-visual event classification performance on AudioSet-20K with different batch sizes. * Results reported in previous work. [†] Results reproduced on our environment.[2]

| Method | Batch Size | Video-to-Audio | | | Audio-to-Video | | | Fine-tuning | | |
|:---|:---|:---:|:---:|:---:|:---:|:---:|:---:|:---:|:---:|:---:|
| | | R@1 | R@5 | R@10 | R@1 | R@5 | R@10 | A | V | A-V |
| CAV-MAE (Gong et al., 2022b) | 108 | 18.8 | 39.5 | 50.1 | 15.1 | 34.0 | 43.0 | 37.7 | 19.8 | 42.0 |
| CAV-MAE-Scale++[†] | 256 | 16.6 | 37.0 | 45.9 | 14.3 | 32.0 | 40.7 | - | 20.0* | - |
| EquiAV | 256 | 23.6 | 46.9 | 56.5 | 23.6 | 44.6 | 54.8 | 40.6 | 22.7 | 43.1 |
| EquiAV | 512 | **27.7** | **51.2** | **60.0** | **25.9** | **50.2** | **56.6** | **42.3** | **24.9** | **44.9** |

**Weight Scales of Loss Function.** We study the effects of varying weight scales, namely $\lambda_{inter}$, $\lambda_{a;intra}$, and $\lambda_{v;intra}$ within the loss function during the pre-training phase of EquiAV. Table E shows that the highest performance is achieved when all weights are equal. We can also observe some insights from this ablation study: first of all, the performance of single-modal fine-tuning is enhanced when the weights of intra-modal loss increase. Conversely, as the proportion of inter-modal loss diminishes, the ability to learn the correspondence across modalities declines, leading to a reduction in retrieval results. Furthermore, when the weight of the intra-modal loss is reduced or applied to only one modality, the performance is degraded. This implies that the single-modal equivariant contrastive learning for both modalities equally contributes to enhancing the generalizability and representation capability of the model.

---

[2]Weights from https://github.com/YuanGongND/cav-mae

Table E: Zero-shot retrieval results on AudioSet and audio-visual event classification performance on AudioSet-20K with varying weights scales of loss function.

| $\lambda_{inter}$ | $\lambda_{a;intra}$ | $\lambda_{v;intra}$ | Zero-shot Retrieval | | Fine-Tuning | | |
|---|---|---|---|---|---|---|---|
| | | | V2A | A2V | A | V | A-V |
| 1 | 1 | 1 | **27.7** | **25.9** | 42.3 | 24.9 | **44.9** |
| 1 | 2 | 2 | 23.5 | 22.4 | **42.4** | **25.2** | 44.7 |
| 1 | 0.5 | 0.5 | 26.9 | 24.7 | 39.4 | 21.6 | 41.9 |
| 1 | 1 | 0 | 22.9 | 22.5 | 38.9 | 19.2 | 40.5 |
| 1 | 0 | 1 | 22.6 | 22.7 | 34.6 | 21.5 | 37.0 |

**Effect of Backbone Encoder Initialization.** We conduct the additional experiment without ImageNet self-supervised initialization. We utilize self-supervised MAE (He et al., 2022) weights for the audio and visual encoders, following previous work. As shown in Table F, our model still outperforms the previous method without the initialization.

Table F: Zero-shot retrieval results on AudioSet and audio-visual event classification performance on AudioSet-20K with and without backbone encoder initialization.

| Method | ImageNet init. | Zero-shot Retrieval | | Fine-Tuning |
|---|---|---|---|---|
| | | V2A | A2V | A-V |
| CAV-MAE | No | - | - | 37.3 |
| CAV-MAE | SSL | 18.8 | 15.1 | 42.0 |
| EquiAV | No | 21.4 | 21.2 | 40.5 |
| EquiAV | SSL | **27.7** | **25.9** | **44.9** |

# D    QUALITATIVE RESULTS

To investigate the capability of EquiAV to capture augmentation-related information, we perform a sound source localization task. We measure the cosine similarity between the mean pooled representation of audio tokens and visual representations of visual patches. We visualize the heatmaps of the original audio-visual input and the two differently augmented versions for comparison. As shown in Figure B, the results demonstrate that EquiAV consistently maintains audio-visual correspondence and accurately identifies the sound source, even in scenarios involving augmented audio-visual inputs. This highlights the strength of our method in not only capturing augmentation-related information within each modality but also in preserving audio-visual correspondence.

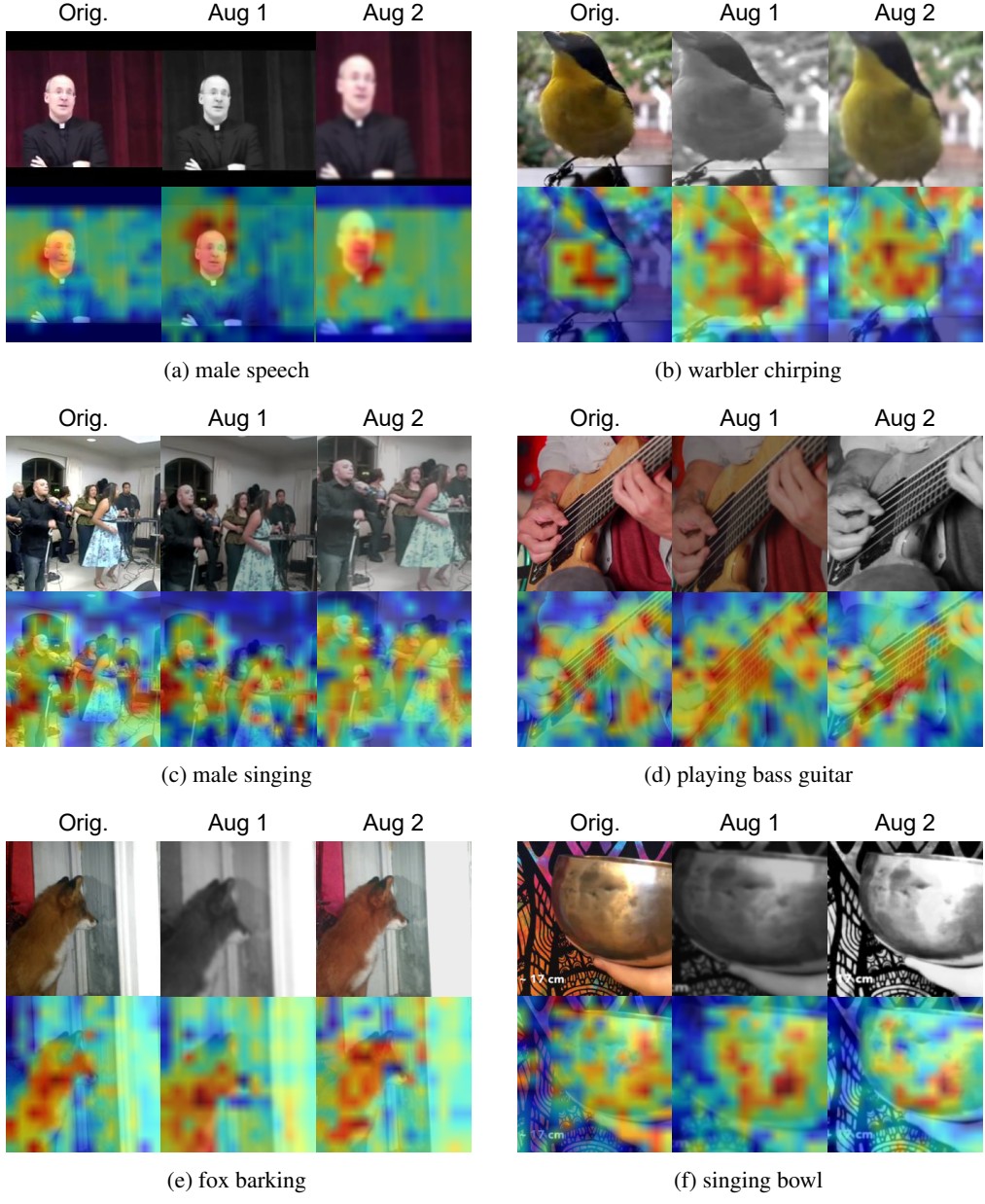

Figure B: Results of visual sound source localization for raw and augmented images. Even for augmented inputs, EquiAV robustly localizes the sound to the visual content.

# E HYPERPARAMETER DETAILS

The hyperparameter settings used in this paper are listed in Table E.

Table G: Hyperparameters used in pre-training and fine-tuning phase.

| Stage | Pre-training | Fine-Tuning | | |
|---|---|---|---|---|
| Dataset | AudioSet-2M | AudioSet-20K | AudioSet-2M | VGGSound |
| Optimizer | AdamW | | | |
| Optimizer momentum | $\beta_1$=0.9, $\beta_2$=0.95 | | | |
| Weight decay | 1e-5 | 1e-5 | 1e-5 | 1e-5 |
| Learning rate scheduler | half-cycle cosine annealing (Loshchilov & Hutter, 2016) | | | |
| Initial learning rate | 1e-6 | 1e-6 | 1e-6 | 1e-6 |
| Peak learning rate | 1e-4 | 1e-4 | 1e-4 | 1e-4 |
| Warm-up epochs | 2 | 1 | 1 | 1 |
| Epochs | 20 | 50 | 50 | 50 |
| Batch size | 64 | 64 | 64 | 64 |
| Class Balancing Weight | No | No | Yes | Yes |
| Mixup | No | Yes | Yes | Yes |
| Loss Function | EquiAV Loss (Eq. 13) | BCE | BCE | CE |
| Temperature ($\tau$) | 0.07 | - | - | - |
| Input Norm Mean | -4.346 | -4.346 | -4.346 | -4.956 |
| Input Norm STD | 4.332 | 4.332 | 4.332 | 4.486 |
| GPUs | 8 A6000 | 8 A5000 | 8 A6000 | 8 A5000 |

## F  ALGORITHM

Algorithm A summarizes EquiAV.

---

**Algorithm A** EquiAV

---

**Input:** audio encoder $f_a$, visual encoder $f_v$,
  audio intra-modal head $g_{a;equ}$, visual intra-modal head $g_{v;equ}$,
  audio augmentation predictor $u_{t_a}$, visual augmentation predictor $u_{t_v}$,
  batch size $N$, temperature $\tau$
  audio augmentation distribution $p_a$, visual augmentation distribution $p_v$
1: **for** sampled mini-batch $\{(x_k^a, x_k^v)\}_{k=1}^N$ **do**
2:     **for all** $k \in \{1, \ldots, N\}$ **do**
3:         sample augmentation instructions $t_a \sim p_a$, $t_v \sim p_v$
            #draw audio features
4:         $z_{2k-1} = g_a(f_a(x_k^a))$
5:         $z_{2k-1}^a = u_{t_a}(g_{a;equ}(f_a(x_k^a)), t_a)$
6:         $z_{2k}^a = g_{a;equ}(f_a(t_a(x_k^a)))$
            #draw video features
7:         $z_{2k} = g_v(f_v(x_k^v))$
8:         $z_{2k-1}^v = u_{t_v}(g_{v;equ}(f_v(x_k^v)), t_v)$
9:         $z_{2k}^v = g_{v;equ}(f_v(t_v(x_k^v)))$
10:     **end for**
11:     **for all** $i \in \{1, \ldots, 2N\}$ and $j \in \{1, \ldots, 2N\}$ **do**
12:         $s_{i,j}^{inter} = \exp\left(\frac{1}{\tau} \cdot \frac{z_i^\top z_j}{\|z_i\|\|z_j\|}\right)$
13:         $s_{i,j}^{a;intra} = \exp\left(\frac{1}{\tau} \cdot \frac{z_i^{a\top} z_j^a}{\|z_i^a\|\|z_j^a\|}\right)$
14:         $s_{i,j}^{v;intra} = \exp\left(\frac{1}{\tau} \cdot \frac{z_i^{v\top} z_j^v}{\|z_i^v\|\|z_j^v\|}\right)$
15:     **end for**
16:     **define** $\ell(i,j)$ **as** $\ell(i,j) = -\log\left(\frac{s_{i,j}}{\sum_{k=1}^{2N} \mathbb{1}_{[k \neq i]} s_{i,k}}\right)$
17:     $\mathcal{L}_{inter} = \frac{1}{2N}\sum_{k=1}^N \left[\ell^{inter}(2k-1, 2k) + \ell^{inter}(2k, 2k-1)\right]$
18:     $\mathcal{L}_{a;intra} = \frac{1}{2N}\sum_{k=1}^N \left[\ell^{a;intra}(2k-1, 2k) + \ell^{a;intra}(2k, 2k-1)\right]$
19:     $\mathcal{L}_{v;intra} = \frac{1}{2N}\sum_{k=1}^N \left[\ell^{v;intra}(2k-1, 2k) + \ell^{v;intra}(2k, 2k-1)\right]$
20:     $\mathcal{L} = \lambda_{inter}\mathcal{L}_{inter} + \lambda_{a;intra}\mathcal{L}_{a;intra} + \lambda_{v;intra}\mathcal{L}_{v;intra}$
21:     update encoders, heads, augmentation predictors to minimize $\mathcal{L}$
22: **end for**

---

