# OpenReview forum: "EquiAV: Single-modal Equivariance Promotes Audio-Visual Contrastive Learning"
_ICLR.cc/2024/Conference — Submitted to ICLR 2024_

### Official Review · Reviewer_4TFL · 2023-10-17

**Soundness:** 3 good
**Presentation:** 3 good
**Contribution:** 1 poor
**Rating:** 3
**Confidence:** 4

**Summary:**

This paper introduces equivariant contrastive learning for intra-model learning in audio-visual self-supervised learning (SSL). It adapts the recently proposed equivariant predictor [1] to predict representations extracted from transformed audio/video from the representations extracted from the untransformed audio/video, respectively in each of the two modalities independently. This replaces the typical invariance loss used in intra-model learning in other audio-visual SSL works e.g. [2], and, as in these previous works, is used in addition to a cross-modal loss, which in this paper enforces invariance as in previous works. Through ablations on invariance vs. equivariance, different augmentations, the changes made to the loss proposed in [1], and different types of initialization for the backbones, the authors reach an optimal training strategy and achieve competitive results on audio/visual/AV classification on AudioSet and VGGSound, and also audio-visual retrieval.



[1] https://openreview.net/pdf?id=eDLwjKmtYFt
[2] https://arxiv.org/pdf/2212.08071v2.pdf

**Strengths:**

The paper is well-written and easy to follow. The narrative and motivation are simple but quite clear and make sense overall. The results seem to confirm the hypotheses that are laid out in the introduction.

The methodology is well explained and does not overcomplicate the description of the losses and learning strategies used in this work/previous works.

Figures 2 and 3 are well-made and highlight the important parts of the method.

The comparison with other works on classifications features a wide range of previous works, table 4 presents a very thorough and welcome ablation study, and the tables are in general clear and well-presented.

The appendix features some further ablations and a lot of details, which lead to some reasonably insightful conclusions and further validate the paper's proposed methodology.

**Weaknesses:**

In short, I think this paper's contribution is not really sufficient, and it is difficult for me to recommend that it be accepted to such a conference in this state, even though it is technically and scientifically sound work.

Basically, what the paper does is apply the intra-model equivariance loss from [1], modify it so that, in the authors' own words, it "differs slightly" from that original loss, and then apply it to each of the modalities (audio and video). The inter-modal loss, which is what makes AVSSL unique, is unchanged. Therefore it is solely applying the methodology of [1] to the modalities of audio and video, and then fine-tuning the hyperparameters as would be done for any uni-modal framework. It is therefore unsurprising, and a direct conclusion of [1], that this would work in some way. Therefore I just don't think there is enough novelty/creativity to this approach to call it truly novel.

We can compare this to, for example, CAV-MAE, which extends AudioMAE but clearly distinguishes itself by proposing a new framework with many new aspects that are unique to their work and exploring how to best model the interactions between audio and video in audio-visual SSL. The same can be said about MAViL, and others - their contributions are more than the direct, naive application of an existing SSL strategy.

The lack of novelty would be acceptable if the results were state-of-the-art by far, but in Table 1 they are outperformed by MAViL in some cases, and in Table 2 they are only compared with a single previous work. As a side note, I don't think it's reasonable to grey out MaVIl as concurrent work in Table 1 - the paper was released on arXiv in Dec. 2022.

Finally, another paper that combines intra-modal and inter-modal losses in self-supervised audio-visual learning (although not in a contrastive way, like MAVil) is RAVEn [2]. Although it is likely not a useful comparison since they only experiment with speech, this is perhaps worth adding to the discussion, especially since their intra-model loss seems to have similar goals to yours (they don't enforce invariance - instead, they leverage a predictor).

Apart from this, I only found a small typo in the title of section C of the appendix: "ADDTIONAL EXPERIMENTS" should be "ADDITIONAL EXPERIMENTS"

[1] https://openreview.net/pdf?id=eDLwjKmtYFt
[2] https://arxiv.org/abs/2212.06246

**Questions:**

Are you planning to release 1. inference code 2. pre-trained models and 3. training code? These would be very welcome, and an important contribution to the audio-visual SSL community.

---

> ### Author Response · Authors · 2023-11-15
> **Author Response to Reviewer 4TFL (1/2)**
>
> Dear Reviewer 4TFL,
>
> Thank you for your constructive and insightful comments. We respond to your questions point-by-point in the following paragraphs.
>
> &nbsp;
>
> ---
>
> > it is solely applying the methodology of [1] to the modalities of audio and video, and then fine-tuning the hyperparameters as would be done for any uni-modal framework. It is therefore unsurprising, and a direct conclusion of [1], that this would work in some way.
> >
>
> As self-supervised learning through invariance has been extended, some studies have shown that applying the concept of equivariance leads to an additional performance boost in various domains such as images and text. Specifically, in the image domain, the concept of equivariance has been applied to SimCLR through developments like E-SSL [2] and EquiMod [1]. Similarly, in the text domain, SimCSE [3] has been advanced with the application of equivariance as seen in DiffCSE [4].
>  These papers show the process of adopting the idea of equivariance for representation learning in each domain. On the other hand, to our knowledge, there is no prior work that incorporates equivariance for audio modality and audio-visual representation learning.
> Along these lines, we aim to design a framework for employing equivariant representation learning in audio-visual representation learning and demonstrate its benefits. However, extending the single-modal equivariant representation learning to multi-modal representation learning is not trivial. Our additional ablation studies (Table 3 in the revised manuscript) reveal that applying equivariance to inter-modal latent spaces is not beneficial and may even hinder the learning of audio-visual correspondences and joint representations. We found that augmentation-related information and audio-visual correspondence need to be learned in separate latent spaces to overcome this challenge. In addition, simply replacing the single-modal invariant representation learning with the equivariant one doesn’t lead to performance improvement. Consequently, we have meticulously designed a framework that maximizes the benefits of equivariance, while avoiding negative impacts on learning. Each component of our framework has been verified to be effective through ablation studies, including:
>
> 1) Loss function (Table 5),
>
> 2) Data augmentations for effective equivariant representation learning (Table 4), and
>
> 3) Projection head and augmentation encoder (Table C in Appendix).
>
> Taken together, all of these factors indicate that our contribution goes beyond simply applying existing techniques to create a new and effective framework for audio-visual representation learning.
>
> &nbsp;
>
> ---
>
> > The lack of novelty would be acceptable if the results were state-of-the-art by far, but in Table 1 they are outperformed by MAViL in some cases, and in Table 2 they are only compared with a single previous work.
> >
>
> There are a couple of key factors to consider in the performance of MAViL:
> - Firstly, MAViL utilizes 8 frames for training, in contrast to the single frame used in CAV-MAE and our work. Although using more frames improves performance, it significantly increases computational costs (MAViL uses 64 gpus vs. EquiAV uses 8 gpus).
> - Secondly, MAViL adopts a two-stage training approach. The first stage involves training the model with masked modeling and contrastive learning techniques, followed by a second stage of retraining using knowledge distillation (KD). It's worth noting that KD could potentially boost performance in other studies too, such as CAV-MAE and Audiovisual MAE, including our own.
> In addition, we have compared our work with various other methods by reporting zero-shot retrieval results on the MSR-VTT dataset, as shown in Table A. The revised manuscript also includes further fine-tuning experiments on UCF101, HMDB51, and ESC50, which can be found in Table B.
>
> &nbsp;
>
> ---
>
> > As a side note, I don't think it's reasonable to grey out MAViL as concurrent work in Table 1 - the paper was released on arXiv in Dec. 2022.
> >
>
> As you correctly noted, MAViL was initially uploaded in December 2022. Our decision to classify it as concurrent work stems from the fact that at the time of our submission in September 2023, we were referencing its revised version from July 2023. Given that MAViL demonstrates performance comparable to our study on the same benchmarks, we included it in our main results following the guideline of covering all relevant papers. If there has been any oversight on our part in this regard, we are open to making any necessary revisions to how MAVil is mentioned in our paper.

---

> > ### Author Response · Authors · 2023-11-15
> > **Author Response to Reviewer 4TFL (2/2)**
> >
> > > Finally, another paper that combines intra-modal and inter-modal losses in self-supervised audio-visual learning (although not in a contrastive way, like MAVil) is RAVEn [5]. Although it is likely not a useful comparison since they only experiment with speech, this is perhaps worth adding to the discussion, especially since their intra-model loss seems to have similar goals to yours (they don't enforce invariance - instead, they leverage a predictor).
> > >
> >
> > As you mentioned, RAVEn [5] also leverages intra-modal loss for audio modality in audio-visual self-supervised learning to encourage the model to retain information from audio input that is absent from the visual counterpart. On the other hand, we apply intra-modal loss for both modalities. The purpose is to make the encoder learn modality-specific information for both modalities, as well as the augmentation-related information which enables the learning of rich representation. By adopting equivariance, we can make the model robust to a strong augmentation which can distort the audio-visual correspondence. We have included RAVEn [5] in our discussion in “Related Work” section in the revised manuscript. Thank you for your suggestion.
> >
> > &nbsp;
> >
> > ---
> >
> > > Apart from this, I only found a small typo in the title of section C of the appendix: "ADDTIONAL EXPERIMENTS" should be "ADDITIONAL EXPERIMENTS”.
> > >
> >
> > We corrected the typo in the revised manuscript. Thank you for your meticulous review.
> >
> > &nbsp;
> >
> > ---
> >
> > > Are you planning to release 1. inference code 2. pre-trained models and 3. training code? These would be very welcome, and an important contribution to the audio-visual SSL community.
> > >
> >
> > Yes. We will release the training & inference codes on GitHub, as well as model checkpoints to allow for reproducibility. We hope that our work will become a contribution to future works on audio-visual representation learning.
> >
> > &nbsp;
> >
> >
> > [1] Alexandre Devillers and Mathieu Lefort. “Equimod: An equivariance module to improve visual instance discrimination”. *In The Eleventh International Conference on Learning Representations*, 2023
> >
> > [2] Dangovski et al., “Equivariant self-supervised learning: Encouraging equivariance in representations”. *In International Conference on Learning Representations*, 2022
> >
> > [3] GAO et al., “Simcse: Simple contrastive learning of sentence embeddings”. *Empirical Methods in Natural Language Processing*, 2021
> >
> > [4] Yung-Sung et al., “DiffCSE: Difference-based contrastive learning for sentence embeddings”. *Annual Conference of the North American Chapter of the Association for Computational Linguistics*, 2022
> >
> > [5] Haliassos et al., “Jointly learning visual and auditory speech representations from raw data”. *In The Eleventh International Conference on Learning Representations*, 2022

---

> ### Comment · Reviewer_4TFL · 2023-11-16
>
> I really appreciate the detailed response.
>
> Regarding novelty, I understand your argument and am not undermining the effort it took to make this configuration work, but still firmly believe that the contribution is not sufficient for a full ICLR paper.
>
> Regarding MAViL, I understand your argument since the results have been updated in the July version but even then I think it's unreasonable to grey it out as concurrent work. I think it's a reasonable comparison.
>
> Thank you for adding RAVEn as a reference, correcting the typo, and agreeing to release the code.
>
> Overall, while I appreciate the authors' detailed responses and new experiments, I don't think it's appropriate to raise the score since I still think the lack of novelty is a major issue.

---

> > ### Author Response · Authors · 2023-11-17
> > **Additional Response to Reviewer 4TFL**
> >
> > Thank you for sharing your opinion.
> >
> > The idea of EquiAV is inspired by EquiMod, but we want to highlight that we successfully extend its work to an audio-visual domain with significant modifications, rather than applying the method of EquiMod as its own.
> >
> > Technical differences compared to EquiMod and our findings can be listed as follows:
> >
> > - **Unlike EquiMod, which uses both invariance and equivariance losses in the single-modal domain, EquiAV applies invariance loss to the inter-modal space and equivariance loss to the intra-modal spaces.** This distinction is crucial since in the inter-modal latent space, it is important to learn the unchanging, significant information shared by audio and visual modalities. Conversely, in the intra-modal latent spaces, utilizing equivariance only is sufficient, as it better captures modality-specific details and learns augmentation-related information. Experimental results in Table 3 demonstrate the effectiveness of this tailored approach.
> > - **Although our equivariance loss has a similar form to the EquiMod loss, they function in a different way during training, leading to a significant performance gap.** As explained in Appendix A of the manuscript, including the positive pair in the denominator results in a relatively larger update for hard positives. This becomes particularly advantageous when stronger augmentations lead to an increased frequency of hard positives. Training with more hard positives, alongside the application of equivariance, substantially enhances the model's capability to comprehend detailed features in single-modal contexts. The intra-modal representation quality plays a pivotal role in understanding and integrating different modalities. Consequently, the semantically rich intra-modal representation promotes effective learning of audio-visual correspondence and audio-visual joint representations.
> > - We have conducted an ablation study on **how to deliver the augmentation information to the augmentation predictor**, as reported in Appendix C.2. In addition to concatenating the input embedding and augmentation vector, we also tried using a 3-layer MLP to encode the augmented vectors to latent vectors and then concatenate it with the input embedding (as shown in Figure A (b)). The results in the first and the second row of Table C show that directly concatenating the augmentation vector with the input embedding gives better results compared to using the additional augmentation encoder. We hypothesize that the augmentation vector is produced through the parameterization of input augmentation and the further encoding of augmentation information by the augmentation encoder might lose some salient information.
> > - To the best of our knowledge, this is the first that the concept of equivariant learning has been applied to multimodality, as well as to the case of the audio domain.
> > In the case of the intra-modal equivariant learning for the audio domain, we didn't just apply existing audio-based augmentations like SpecAug [6] but adopted visual-based augmentations (e.g. random crop, horizontal flip, etc.) as well. Experimental results conclusively demonstrate that these visual-based augmentations significantly enhance learning both audio representation (shown in Table 1, 4, and B)  and audio-visual correspondence (shown in Table 1, 2, 4, and A). **This is definitely our contribution entirely absent in prior works.**
> >
> > As also mentioned in the global response, we understand the concerns regarding novelty. However, we believe that extending a method proven effective in one domain to another, thereby making it applicable in a new context is also an essential area of AI research.
> >
> > Thank you again for giving your comments, and we are always open to any constructive feedback that can improve our paper for the audio-visual SSL community.
> >
> > &nbsp;
> >
> > [6] Park et al., “Specaugment: A simple data augmentation method for automatic speech recognition”. *In Interspeech*, pp. 2613–2617, 2019

---

> > > ### Comment · Reviewer_4TFL · 2023-11-20
> > >
> > > I acknowledge, understand, and appreciate your response, but respectfully still stand by my earlier comments.

---

### Official Review · Reviewer_A7KE · 2023-10-28

**Soundness:** 3 good
**Presentation:** 3 good
**Contribution:** 3 good
**Rating:** 6
**Confidence:** 3

**Summary:**

This work introduces EquiAV, a framework to incorporate data augmentation in audio-visual representation learning. Modality specific information is learned in a separate space from the audio-visual correspondence space. Furthermore, the modality-specific representations are learned such that they are equivariant to data augmentations. This work demonstrated the benefits of the proposed framework through downstream tasks of event classification and zero-shot retrieval. In a addition to this, an ablation study is also provided to show model robusteness to augmentation along with the proposed approach's ability to maximize the benefits of augmentation itself.

**Strengths:**

Originality :

The work is somewhat original as it combines a contrastive learning framework with an equivariance framework that was extended to a multimodal formulation from a single modality formulation.

Quality :

This work creates a well structured framework to introduce data augmentations as part of a contrastive loss based learning of audio-visual representations. The proposed approach is supported by ablation studies and explorations on several downstream tasks. However, there are some questions that come up.

Clarity :

This work is somewhat clear although there are some parts (such as those mentioned under Questions) that could be made more clear.


Significance:

Data augmentation is one of the ways to not just improve the performance of models but also to increase their robustness. To this end, the proposed work presents a step forward in the context of audio-visual (and potentially general multimodal) representation learning.

**Weaknesses:**

Although the work is well structured, some of the decisions/formulations are not fully explained (as mentioned in the Questions section). This work can also benefit by providing additional evidence to see of its claims through different downstream tasks and ablation studies (as indicated in the Questions section)

**Questions:**

If the loss is formulation (1) is contrastive in nature then the $\mathcal{L}$ should not just output the dissimilarity as the optimization minimizes $\mathcal{L}$ and contrastive loss is about maximizing similarity of aligned data and minimizing similarity of non-aligned data. I assume that the author's meant a formulation of contrastive loss and not just a dissimilarity measuring mechanism. Please clarify.

Conceptually, the augmented and non-augmented variants can be considered as two different modalities. Therefore, standard contrastive loss setup (as used in Eq 11) can be used. Is there a specific reason for using the Eq 9 formulation.

Furthermore, the reasoning behind inclusion of  'positive pair ...in the denominator ' is not clear. The appendix does seem to allude to the different weights during gradient calculations but does not elaborate on why those specific weights should necessarily be a factor. Table 5 does give results that demonstrate that it can be important but there is not an explanation as to why it is so.

There are two types of features for each modality, one from the head corresponding to the inter-modal space and the other from the intra-modal space. Perhaps I missed it, but it is not apparent which features were used in the experiments.

According to the manuscript, Table 4 implies 'that by utilizing equivariant intra-modal representation learning, our model is able to capture augmentation-related information within each modality'. It would be good to dig deeper into this facet perhaps through heatmaps to show stronger evidence as Table 4 results show that the proposed setup is good for the given downstream tasks which is perhaps not enough evidence to strongly claim the models ability to capture augmentation related information.

As there are individual modality branches (and multiple types of modality features) available it would be beneficial to have comparisons with other (non-augmentation focused) baselines on common tasks such as unimodal action recognition (such as on UCF and HMDB) and sound classification (such as on ESC) that are often explored in works that explore audio-visual representation learning.

---

> ### Author Response · Authors · 2023-11-15
> **Author Response to Reviewer A7KE (1/2)**
>
> Dear Reviewer A7KE,
>
> Thank you for your valuable suggestions to improve our paper. Please see the point-by-point responses below.
>
> &nbsp;
>
> > If the loss is formulation (1) is contrastive in nature then the $\mathcal{L}$ should not just output the dissimilarity as the optimization minimizes $\mathcal{L}$ and contrastive loss is about maximizing similarity of aligned data and minimizing similarity of non-aligned data. I assume that the author's meant a formulation of contrastive loss and not just a dissimilarity measuring mechanism. Please clarify.
> >
>
> First, it's correct to consider $\mathcal{L}$ as a contrastive loss, as you've mentioned. In our work, we aimed to establish a more general formulation for representation learning. Specifically, we formulate it as maximizing the similarity between pairs of inputs, which is equivalent to minimizing their dissimilarity.
> The four equations described in Section 2 of our manuscript are designed to be applicable not only to contrastive learning but also to other forms of representation learning. Furthermore, these formulations are adaptable to a variety of techniques used to prevent collapse in Siamese networks, such as the use of 'stop gradient', illustrating their broad applicability in recent advancements in the field.
>
> &nbsp;
>
> ---
>
> > Conceptually, the augmented and non-augmented variants can be considered as two different modalities. Therefore, standard contrastive loss setup (as used in Eq 11) can be used. Is there a specific reason for using the Eq 9 formulation.
> >
>
> As you pointed out, they can be treated as two different modalities, which would allow the use of Eq 11. However, this method halves the number of negative pairs (from 2N-2 to N-1), which is similar to decreasing the batch size. Such a reduction in negative pairs can have a negative impact on performance, particularly in the context of contrastive learning. While Eq 11 may have its advantages when using different encoders for significantly different spaces, like in video-audio or image-text pair, we can conclude that Eq 9 is more suitable for intra-modal training.
>
> &nbsp;
>
> ---
>
> > Furthermore, the reasoning behind inclusion of 'positive pair ...in the denominator ' is not clear. The appendix does seem to allude to the different weights during gradient calculations but does not elaborate on why those specific weights should necessarily be a factor. Table 5 does give results that demonstrate that it can be important but there is not an explanation as to why it is so.
> >
>
> As explained in Appendix A of the manuscript, including the positive pair in the denominator results in a relatively larger update for hard positives. This becomes particularly advantageous when stronger augmentations lead to an increased frequency of hard positives. Training with more hard positives, alongside the application of equivariance, substantially enhances the model's capability to comprehend detailed features in single-modal contexts. The intra-modal representation quality plays a pivotal role in understanding and integrating different modalities. Consequently, the semantically rich intra-modal representation promotes effective learning of audio-visual correspondence and audio-visual joint representations. We will add a detailed analysis in Appendix A.
>
> &nbsp;
>
> ---
>
> > There are two types of features for each modality, one from the head corresponding to the inter-modal space and the other from the intra-modal space. Perhaps I missed it, but it is not apparent which features were used in the experiments.
> >
>
> We employed features projected into the inter-modal space for all downstream tasks, and Table C in Appendix C.2 demonstrates which features yield the best performance. As you mentioned, we conducted experiments using features projected into the intra-modal space, as well as features that were processed solely through the encoder without any heads. The results indicate that features projected into the inter-modal space exhibit the best performance.
>
> &nbsp;
>
> ---
>
> > It would be good to dig deeper into this facet perhaps through heatmaps to show stronger evidence as Table 4 results show that the proposed setup is good for the given downstream tasks which is perhaps not enough evidence to strongly claim the models ability to capture augmentation related information.
> >
>
> In response to your suggestion, we have further demonstrated the capability of EquiAV to capture augmentation-related information by conducting a visual sound source localization task. The qualitative results are detailed in Figure B in Appendix D of the revised manuscript. The results demonstrate that EquiAV consistently maintains audio-visual correspondence and accurately identifies the sound source, even in scenarios involving augmented audio-visual inputs. This highlights the strength of our method in not only capturing augmentation-related information within each modality but also in preserving audio-visual correspondence.

---

> > ### Author Response · Authors · 2023-11-15
> > **Author Response to Reviewer A7KE (2/2)**
> >
> > > As there are individual modality branches (and multiple types of modality features) available it would be beneficial to have comparisons with other (non-augmentation focused) baselines on common tasks such as unimodal action recognition (such as on UCF and HMDB) and sound classification (such as on ESC) that are often explored in works that explore audio-visual representation learning.
> > >
> >
> > Following your advice, we conducted experiments to demonstrate the effectiveness of our method in single-modal downstream tasks. For the video-only task, we employed UCF101 and HMDB51 datasets, and for the audio-only task, the ESC-50 dataset. Since many audio-visual based papers do not report the results of such experiments, we directly reproduced the model with publicly available weights.
> > As shown below, our method showed superior performance in the single-modal downstream tasks compared to previous audio-visual representation learning methods. We have reported detailed results including audio-only methods in Table B of the revised manuscript.
> >
> > | Model | video-only | video-only | audio-only |
> > | --- | --- | --- | --- |
> > |  | UCF101 | HMDB51 | ESC-50 |
> > | CAV-MAE [1] | 75.9 | 43.5 | 84.0 |
> > | MAViL [2] | - | - | 94.4 |
> > | EquiAV (ours) | **87.0** | **65.5** | **96.0** |
> >
> > &nbsp;
> >
> > [1] Gong et al., “Contrastive audio-visual masked autoencoder”. *In The Eleventh International Conference on Learning Representations*, 2022
> >
> > [2] Huang et al., “Mavil: Masked audio-video learners”. *arXiv preprint arXiv:2212.08071*, 2022

---

> > > ### Comment · Reviewer_A7KE · 2023-11-22
> > >
> > > Thanks for the author response and clarifications

---

### Official Review · Reviewer_eQDc · 2023-10-31

**Soundness:** 2 fair
**Presentation:** 3 good
**Contribution:** 2 fair
**Rating:** 3
**Confidence:** 5

**Summary:**

The EquiAV framework proposed in this paper aims to improve audio-visual self-supervised learning by finding a strategy for utilizing data augmentation that maximizes the benefits of the model while maintaining robustness to substantial augmentation. The approach combines single-modal equivariant contrastive learning and audio-visual contrastive learning to learn audio-visual correspondence and modality-specific representations separately. The paper also compared various techniques employed in audio-visual representation learning. The EquiAV ensures that diverse augmentations applied to both audio and visual modalities benefit the model. Experimental results demonstrate that the EquiAV approach outperforms existing state-of-the-art methods in audio-visual event classification and zero-shot audio-visual retrieval tasks. Extensive ablation studies are conducted to demonstrate the effectiveness of the proposed method in learning audio-visual correspondence and enhancing representation capability.

**Strengths:**

- The author uses detailed comparative experiments and ablation experiments to prove the effectiveness and advancement of the EquiAV framework using single-modal equivariant representation learning. The article is also logically clear and uses reasonable diagrams to explain the relevant content clearly.
- In the experiment, EquiAV showed impressive results and also provided the best settings for Audio-visual data augmentation as a reference for subsequent research.

**Weaknesses:**

- The novelty of the article is limited. The article only applies the single-modal equivariant representation learning that has been proven effective to the A-V learning task and does not try to solve particular problems in this field (for example, compared with the Text-audio, Text-vision field, what specific difficulties can this method solve?)
- Although the author tried to compare different pre-training methods, he did not clearly explain the advantages of equivariant representation learning.
- The author lacks design details for augmentation predictors in the article. How does it work? Why can it make the framework achieve better results? What are the specific settings? Whether to only perform linear transformations on original input embeddings and augmentation vectors
- The author uses InvAV as the baseline but does not give a reference to this solution.
- Tables 1 and 5 appear in the wrong chapter positions, and the layout of the article needs to be carefully revised.

**Questions:**

- The specific details of Figure 1 are missing. What is the specific meaning of augmentation level? What specific settings were used to draw this figure?
- In Table 1, the performance of MAViL is better than EquiAV. What is the specific reason?
- Applying intra-modal representations equivariant to the augmentations has been proven effective in previous research. What is the novelty of this article?

---

> ### Author Response · Authors · 2023-11-15
> **Author Response to Reviewer eQDc (1/2)**
>
> Dear Reviewer eQDc,
>
> We appreciate your valuable comments. We list the responses to the concerns below.
>
> &nbsp;
>
> ---
>
> > Applying intra-modal representations equivariant to the augmentations has been proven effective in previous research. What is the novelty of this article?
> >
>
> As self-supervised learning through invariance has been extended, some studies have shown that applying the concept of equivariance leads to an additional performance boost in various domains such as images and text. Specifically, in the image domain, the concept of equivariance has been applied to SimCLR through developments like E-SSL[2] and EquiMod[1]. Similarly, in the text domain, SimCSE[3] has been advanced with the application of equivariance as seen in DiffCSE[4].
> These papers show the process of adopting the idea of equivariance for representation learning in each domain. On the other hand, to our knowledge, there is no prior work that incorporates equivariance for audio modality and audio-visual representation learning.
> Along these lines, we aim to design a framework for employing equivariant representation learning in audio-visual representation learning and demonstrate its benefits. However, extending the single-modal equivariant representation learning to multi-modal representation learning is not trivial. Our additional ablation studies (Table 3 in the revised manuscript) reveal that applying equivariance to inter-modal latent spaces is not beneficial and may even hinder the learning of audio-visual correspondences and joint representations. We found that augmentation-related information and audio-visual correspondence need to be learned in separate latent spaces to overcome this challenge. In addition, simply replacing the single-modal invariant representation learning with the equivariant one doesn’t lead to performance improvement. Consequently, we have meticulously designed a framework that maximizes the benefits of equivariance, while avoiding negative impacts on learning. Each component of our framework has been verified to be effective through ablation studies, including:
>
> 1) Loss function (Table 5),
>
> 2) Data augmentations for effective equivariant representation learning (Table 4), and
>
> 3) Projection head and augmentation encoder (Table C in Appendix).
>
> Taken together, all of these factors indicate that our contribution goes beyond simply applying existing techniques to create a new and effective framework for audio-visual representation learning.
>
> &nbsp;
>
> ---
>
> > The specific details of Figure 1 are missing. What is the specific meaning of augmentation level? What specific settings were used to draw this figure?
> >
>
> We drew Figure 1 based on the ablation results in Table 4. The term "augmentation level" refers to the variety and strength of applied augmentations, which correspond to the augmentation settings described in each row of Table 4. For example, augmentation level 0 corresponds to the first row, 1 corresponds to the second row, and so on. We added the notion “For detailed augmentation settings, refer to Table 4" to the caption of Figure 1. If there is a better way to describe Figure 1, we would appreciate it.
>
> &nbsp;
>
> ---
>
> > In Table 1, the performance of MAViL is better than EquiAV. What is the specific reason?
> >
>
> There are a couple of key factors to consider in the performance of MAViL:
> - Firstly, MAViL utilizes 8 frames for training, in contrast to the single frame used in CAV-MAE and our work. Although using more frames improves performance, it significantly increases computational costs (MAViL uses 64 gpus vs. EquiAV uses 8 gpus).
> - Secondly, MAViL adopts a two-stage training approach. The first stage involves training the model with masked modeling and contrastive learning techniques, followed by a second stage of retraining using knowledge distillation (KD). It's worth noting that KD could potentially boost performance in other studies too, such as CAV-MAE and Audiovisual MAE, including our own.
> Please note that MAViL is our concurrent work and we included it as grayed out in our main results following the guideline of covering all relevant papers.
>
> &nbsp;
>
> ---
>
> > Although the author tried to compare different pre-training methods, he did not clearly explain the advantages of equivariant representation learning.
> >
>
> As demonstrated in Table 4, learning representations equivariant rather than invariant to augmentations allows for more robust learning of the semantics of inputs without compromising them. Consequently, this leads to better learning of audio-visual correspondence and joint representation. We also provide additional supporting evidence through an ablation study (Table 3 in the revised manuscript) for a more comprehensive understanding of our findings.

---

> > ### Author Response · Authors · 2023-11-15
> > **Author Response to Reviewer eQDc (2/2)**
> >
> > > The author lacks design details for augmentation predictors in the article.
> > >
> >
> > The role of the augmentation predictor is to estimate the displacement in the latent space caused by input augmentation. We have conducted an ablation study on how to deliver the augmentation information to the augmentation predictor, as reported in Appendix C.2. In addition to concatenating the input embedding and augmentation vector, we also tried using a 3-layer MLP to encode the augmented vectors to latent vectors and then concatenate it with the input embedding (as shown in Figure A (b)). The results in the first and the second row of Table C (in the revised manuscript) show that directly concatenating the augmentation vector with the input embedding gives better results compared to using the additional augmentation encoder. The augmentation vector is produced through the parameterization of input augmentation and the further encoding of augmentation information by the augmentation encoder might lose some salient information.
> >
> > &nbsp;
> >
> > ---
> >
> > > The author uses InvAV as the baseline but does not give a reference to this solution.
> > >
> >
> > InvAV can be considered as adding intra-modal contrastive loss to CAV in CAV-MAE, or the first pre-training stage in MAViL without masked modeling. There exist works that employ intra-modal contrastive loss for audio-visual representation learning, but InvAV is not designed in exactly the same way as those works. It is just a baseline for comparison with our method.
> >
> > &nbsp;
> >
> > ---
> >
> > > Tables 1 and 5 appear in the wrong chapter positions, and the layout of the article needs to be carefully revised.
> > >
> >
> > Following your advice, we revised the layout of the article so that the tables appear in the right positions of the corresponding chapters. Thank you for giving a delicate review.
> >
> >
> > &nbsp;
> >
> >
> > [1] Alexandre Devillers and Mathieu Lefort. “Equimod: An equivariance module to improve visual instance discrimination”. *In The Eleventh International Conference on Learning Representations*, 2023
> >
> > [2] Dangovski et al., “Equivariant self-supervised learning: Encouraging equivariance in representations”. *In International Conference on Learning Representations*, 2022
> >
> > [3] GAO et al., “Simcse: Simple contrastive learning of sentence embeddings”. *Empirical Methods in Natural Language Processing*, 2021
> >
> > [4] Yung-Sung et al., “DiffCSE: Difference-based contrastive learning for sentence embeddings”. *Annual Conference of the North American Chapter of the Association for Computational Linguistics*, 2022

---

### Official Review · Reviewer_3ZPL · 2023-11-03

**Soundness:** 3 good
**Presentation:** 3 good
**Contribution:** 1 poor
**Rating:** 3
**Confidence:** 2

**Summary:**

In this paper, the authors introduce EquiAV, a new framework that integrates single-modal equivariant contrastive learning with audio-visual contrastive learning. In the proposed framework, audio-visual correspondence and rich modality-specific representations are learned in separate latent spaces.  Extensive ablation studies verify that EquiAV outperforms the existing audio-visual self-supervised learning methods on audio-visual event classification and zero-shot audio-visual retrieval tasks.

**Strengths:**

The authors extend single-modal equivariant representation learning to the audio-visual domain, achieving better performance than previous audio-visual self-supervised learning methods.

**Weaknesses:**

+ Novelty. The technical contribution of the proposed method is relatively limited. It extends the existing EQUIMOD [1] method to audio and visual modalities, applying it to audio-visual self-supervised learning. I do not see any specific contributions of this work to the audio-visual learning field. The intra loss was applied to the two modalities separately, and I do not believe that this work contributes any new insights into cross-modal modality. Simply applying a state-of-the-art approach to a new application does not necessarily result in new significant contributions.

+ Writing. Some statements about the key contributions are vague. For example, the authors state that "employing augmentations in multi-modal contrastive learning requires careful consideration, as augmentations can severely distort the inter-modal correspondence." It would be helpful to provide some specific examples to illustrate how augmentations can distort the inter-modal correspondence. Additionally, "the equivariance loss term in the proposed framework differs slightly from the single-modal equivariant self-supervised learning (Devillers & Lefort, 2023). The key distinction is whether or not the similarity of the positive pair is included in the denominator of the loss term." However, it is unclear why the positive pair is included in the denominator. Finally, the authors state that "to model the displacement in the latent space caused by data augmentations, augmentation predictors take as input the concatenation of the original input embeddings and the augmentation vectors, and output equivariant embeddings." It would be helpful to describe more specifically what kinds of displacements these predictors can model and why concatenating the input embedding and augmentation vector is helpful.

+ Experiment. What if we applied Equimod to audio and visual data directly without the positive term in the loss?

[1] Alexandre Devillers and Mathieu Lefort. Equimod: An equivariance module to improve visual instance discrimination. ICLR, 2023.

**Questions:**

Please address questions in Weaknesses.

---

> ### Author Response · Authors · 2023-11-15
> **Author Response to Reviewer 3ZPL (1/2)**
>
> Dear Reviewer 3ZPL,
>
> Thank you for giving us valuable comments. We present responses to the raised concerns and questions below.
>
> &nbsp;
>
> > It extends the existing EQUIMOD [1] method to audio and visual modalities, applying it to audio-visual self-supervised learning. I do not see any specific contributions of this work to the audio-visual learning field.
> >
> As self-supervised learning through invariance has been extended, some studies have shown that applying the concept of equivariance leads to an additional performance boost in various domains such as images and text. Specifically, in the image domain, the concept of equivariance has been applied to SimCLR through developments like E-SSL [2] and EquiMod [1]. Similarly, in the text domain, SimCSE [3] has been advanced with the application of equivariance as seen in DiffCSE [4].
> These papers show the process of adopting the idea of equivariance for representation learning in each domain. On the other hand, to our knowledge, there is no prior work that incorporates equivariance for audio modality and audio-visual representation learning.
> Along these lines, we aim to design a framework for employing equivariant representation learning in audio-visual representation learning and demonstrate its benefits. However, extending the single-modal equivariant representation learning to multi-modal representation learning is not trivial.  Our additional ablation studies (Table 3 in the revised manuscript) reveal that applying equivariance to inter-modal latent spaces is not beneficial and may even hinder the learning of audio-visual correspondences and joint representations. We found that augmentation-related information and audio-visual correspondence need to be learned in separate latent spaces to overcome this challenge. In addition, simply replacing the single-modal invariant representation learning with the equivariant one doesn’t lead to performance improvement. Consequently, we have meticulously designed a framework that maximizes the benefits of equivariance, while avoiding negative impacts on learning. Each component of our framework has been verified to be effective through ablation studies, including:
>
> 1) Loss function (Table 5),
>
> 2) Data augmentations for effective equivariant representation learning (Table 4), and
>
> 3) Projection head and augmentation encoder (Table C in Appendix).
>
> Taken together, all of these factors indicate that our contribution goes beyond simply applying existing techniques to create a new and effective framework for audio-visual representation learning.
>
> &nbsp;
>
> ---
> > It would be helpful to provide some specific examples to illustrate how augmentations can distort the inter-modal correspondence.
> >
>
> For a more straightforward understanding, we came up with a case where data augmentation causes distortion of the inter-modal correspondence. Let’s suppose that given an audio-visual pair of a barking dog, the dog disappears due to augmentation (e.g. random cropping). When using intra-modal invariant learning, the features of the original image (i.e. the dog is barking) and the augmented one (i.e. the dog disappears) are forced to become closer in the latent space. This will give the wrong signal to the model, adversely affecting the learning of correct inter-modal correspondence (i.e. the dog and the barking sound). On the other hand, in the context of equivariant learning, the embedding of the original image is transformed by the augmentation predictor and made closer to the embeddings of the augmented image in the intra-modal latent space. Then, we utilize original inputs (i.e. dog and the barking sound) that retain the semantics of the two modalities in the inter-modal latent space while ensuring that the embeddings of the augmented image and the original audio will not become directly close. This insight allows equivariance to take advantage of the effect of strong data augmentation in audio-visual representation learning.
>
> &nbsp;
>
> ---
>
> > it is unclear why the positive pair is included in the denominator.
> >
>
> As explained in Appendix A of the manuscript, including the positive pair in the denominator results in a relatively larger update for hard positives. This becomes particularly advantageous when stronger augmentations lead to an increased frequency of hard positives. Training with more hard positives, alongside the application of equivariance, substantially enhances the model's capability to comprehend detailed features in single-modal contexts. The intra-modal representation quality plays a pivotal role in understanding and integrating different modalities. Consequently, the semantically rich intra-modal representation promotes effective learning of audio-visual correspondence and audio-visual joint representations. We will add a detailed analysis in Appendix A.

---

> > ### Author Response · Authors · 2023-11-15
> > **Author Response to Reviewer 3ZPL (2/2)**
> >
> > > It would be helpful to describe more specifically what kinds of displacements these predictors can model and why concatenating the input embedding and augmentation vector is helpful.
> > >
> >
> > If an original input and its augmented version are projected into the latent space, they will be projected in the neighborhood but at some distance. The role of the augmentation predictor is to estimate the displacement in the latent space caused by input augmentation. We have conducted an ablation study on how to deliver the augmentation information to the augmentation predictor, as reported in Appendix C.2. In addition to concatenating the input embedding and augmentation vector, we also tried using a 3-layer MLP to encode the augmented vectors to latent vectors and then concatenate it with the input embedding (as shown in Figure A (b)). The results in the first and the second row of Table C show that directly concatenating the augmentation vector with the input embedding gives better results compared to using the additional augmentation encoder. We hypothesize that the augmentation vector is produced through the parameterization of input augmentation and the further encoding of augmentation information by the augmentation encoder might lose some salient information.
> >
> > &nbsp;
> >
> > ---
> > > What if we applied Equimod to audio and visual data directly without the positive term in the loss?
> > >
> >
> > We have indeed explored this scenario, and the results are presented in Table 5 of our paper. The results indicate that using the loss function employed in Equimod leads to suboptimal performance in downstream tasks.  For a more in-depth analysis of these results, please refer to the discussion provided in Appendix A.
> >
> > &nbsp;
> > ---
> > [1] Alexandre Devillers and Mathieu Lefort. “Equimod: An equivariance module to improve visual instance discrimination”. *In The Eleventh International Conference on Learning Representations*, 2023
> >
> > [2] Dangovski et al., “Equivariant self-supervised learning: Encouraging equivariance in representations”. *In International Conference on Learning Representations*, 2022
> >
> > [3] GAO et al., “Simcse: Simple contrastive learning of sentence embeddings”. *Empirical Methods in Natural Language Processing*, 2021
> >
> > [4] Yung-Sung et al., “DiffCSE: Difference-based contrastive learning for sentence embeddings”. *Annual Conference of the North American Chapter of the Association for Computational Linguistics*, 2022

---

### Author Response · Authors · 2023-11-15
**General Response to Reviewers**

Dear Reviewers, we appreciate your thorough reviews and the insightful feedback provided on our manuscript.

We understand that some concerns were raised regarding the novelty of our proposed method and the clarity of its contribution. While we recognize that our architectural innovations may not be fundamentally novel, we believe that scalability to other domains is a significant contribution to the AI research community. For instance, AST [1] successfully introduced the vision transformer to the audio domain, and the Audio-MAE [2] extended the masked image modeling technique of MAE to the audio domain. Our research focuses on this scalability, and the novelty of our work lies in the application of equivariant representation learning principles to the audio-visual domain.

Our findings revealed that directly applying equivariance to inter-modal latent spaces hinders the learning of audio-visual correspondences. To address this issue, we proposed a framework that separates the learning of augmentation-related information and audio-visual correspondence in distinct latent spaces. This approach not only preserves the benefits of equivariance but also overcomes the potential negative impacts on joint representation learning.

Through meticulous design and extensive ablation studies, each component of our framework has been proved effective. We believe that our work provides a substantial foundation for future advancements in audio-visual representation learning methods that incorporate the principles of equivariance.

We respectfully request the reviewers to consider these points in reevaluating the impact and value of our work. We thank you again for your constructive review and valuable insights.

Here is the list of revisions incorporated into our paper:
1. Modification of the caption for Figure 1.
2. Extra experiments and explanation of ablations in Table 3.
3. Adding a reference in Related Works.
4. Detailed explanation of the impact of hard positive pairs in the loss function in Appendix A.
5. Single-modal downstream tasks in Appendix C.1.
6. Qualitative results in Appendix D.

&nbsp;

[1] Yuan Gong, Yu-An Chung, and James Glass. Ast: Audio spectrogram transformer. In Interspeech, pp. 571–575, 2021

[2] Po-Yao Huang, Hu Xu, Juncheng B Li, Alexei Baevski, Michael Auli, Wojciech Galuba, Florian Metze, and Christoph Feichtenhofer. Masked autoencoders that listen. In Alice H. Oh, Alekh Agar- wal, Danielle Belgrave, and Kyunghyun Cho (eds.), Advances in Neural Information Processing Systems, 2022

---

### Meta-Review · Area_Chair_b4qk · 2023-11-26

**Metareview:**

This paper introduces EquiAV,  a contrastive learning framework that integrates single-modal equivariant contrastive learning for audio-visual self-supervised learning (SSL). EquiAV adapts a recently proposed single-modal equivariant contrastive learning, EquiMod, to audio-visual representations learning. EquiAV separates the learning of single-modal, augmentation-related information for audio/visual modality from the learning of audio-visual correspondence in distinct latent spaces, in order to mitigate the potential negative impacts when learning representation from multiple modalities jointly. Extensive ablation shows EquiAV achieves strong performance on audio-visual event classification and zero-shot audio-visual retrieval tasks, which indicate the effectiveness of proposed method.

Strength:
 - The paper is well-written and easy to follow. Diagrams and figures are well-made. The narrative and motivation are simple but quite clear and make sense overall.
 - The paper conducts detailed experiments and ablation study to support the effectiveness of proposed EquiAV framework. The results seem to confirm the hypotheses that are laid out in the introduction.
 - The extension of single-modal equivariant representation learning to the audio-visual domain yields interesting and competitive results.

Weakness:
 - Most reviewers raised the concerns of lacking of novelty and contribution since reviewers viewed EquiAV as an multimodal version of EquiMod, while the authors argued the main contribution is in the detail and all the heavy lifting to make single modal approach works for multimodal, such as the separation of single- and cross- modal learning. I agree with the reviewers and also share the concerns that the paper might be of limited interests to ICLR community. To me, making reasonable adjustment to existing technique for new tasks/domains/settings is a basic requirement. However, in ICLR, we are looking for works more than that, and what making a work impressive is usually the innovation of the methodolgy or problem formulation, the cleaness of the design, the insight drawing from the experiment results, etc.
 - As reviewer 4TFL suggests, sometimes signficant SOTA results trump the previous weakness. However, combining the authors' arguement that non-trivial efforts have been put to make EquiMod work at multimodal setting, and EquiAV doesn't outperform consistently to some previous works (e.g., MAViL, the authors argue MAViL as concurrent work, which I disagree since MAViL has been on arXiv since Dec'22), I am worried the improvement might not be strong enough to justify a general interests for ICLR community.

**Justification For Why Not Higher Score:**

As mentioned in the weakness of this paper, this paper works on adapting a signle modal contrastive learning approach (EquiMod) for audio-visual SSL tasks. The novelty is limited and results are not strong enough to justify a general interests for ICLR community.

**Justification For Why Not Lower Score:**

N/A

---

### Decision · Program_Chairs · 2024-01-16

Reject